# Natural transformation allows transfer of SCC*mec*-mediated methicillin resistance in *Staphylococcus aureus* biofilms

Mais Maree [1,7 ✉], Le Thuy Thi Nguyen[2,3,7], Ryosuke L. Ohniwa [4,5,7], Masato Higashide[6], Tarek Msadek [4,8 ✉] & Kazuya Morikawa [5,8 ✉]

SCC*mec* is a large mobile genetic element that includes the *mecA* gene and confers resistance to β-lactam antibiotics in methicillin-resistant *Staphylococcus aureus* (MRSA). There is evidence that SCC*mec* disseminates among staphylococci, but the transfer mechanisms are unclear. Here, we show that two-component systems mediate the upregulation of natural competence genes in *S. aureus* under biofilm growth conditions, and this enhances the efficiency of natural transformation. We observe SCC*mec* transfer via natural transformation from MRSA, and from methicillin-resistant coagulase-negative staphylococci, to methicillin-sensitive *S. aureus*. The process requires the SCC*mec* recombinase genes *ccrAB*, and the stability of the transferred SCC*mec* varies depending on SCC*mec* types and recipients. Our results suggest that natural transformation plays a role in the transfer of SCC*mec* and possibly other mobile genetic elements in *S. aureus* biofilms.

[1] Graduate School of Comprehensive Human Sciences, University of Tsukuba, Tsukuba, Japan. [2] Human Biology Program, School of Integrative and Global Majors, University of Tsukuba, Tsukuba, Japan. [3] Biotechnology Centre of Ho Chi Minh City, District 12, HCM City, Vietnam. [4] Institut Pasteur, Université Paris Cité, CNRS UMR6047, Biology of Gram-Positive Pathogens, Department of Microbiology, F-75015 Paris, France. [5] Division of Biomedical Science, Faculty of Medicine, University of Tsukuba, Tsukuba, Japan. [6] Kotobiken Medical Laboratories, Inc., Kamiyokoba, Tsukuba, Japan. [7] These authors contributed equally: Mais Maree, Le Thuy Thi Nguyen, Ryosuke L. Ohniwa [8] These authors jointly supervised this work: Tarek Msadek, Kazuya Morikawa. ✉email: maismaree1@gmail.com; tarek.msadek@pasteur.fr; morikawa.kazuya.ga@u.tsukuba.ac.jp

*S*taphylococcus aureus, a Gram-positive bacterium belonging to the *Firmicutes* phylum, is present in the nasal cavities of about 30 percent of the human population. *S. aureus* carriers are normally asymptomatic but opportunistic infections, ranging from minor skin abscesses to severe diseases (such as pneumonia, osteomyelitis, or toxic shock syndrome), occur. Immunocompromised hosts are vulnerable, but the spread of community-associated infections by highly virulent *S. aureus* has also been reported[1].

Antibiotic resistance is the most notorious feature of this pathogen, particularly methicillin-resistant *S. aureus* (MRSA). MRSA is the major cause of nosocomial infections (healthcare-associated MRSA, HA-MRSA) and is also associated with healthy individuals (community-associated MRSA, CA-MRSA) and livestock (livestock-associated MRSA, LA-MRSA), posing a global health burden[2–5]. The percentage of MRSA among *S. aureus* isolates from inpatients differs between countries (Vietnam 73%, United States 45%, Japan 41%, and North Europe 1%)[6], raising concerns in clinics, care homes, and other areas with high densities of immunocompromised individuals.

The global spread of this major human pathogen is essentially due to a sophisticated arsenal of virulence factors and antibiotic-resistance genes, many of them located on mobile genetic elements (MGEs) such as plasmids, prophages, transposons, pathogenicity islands, insertion sequences, and the staphylococcal cassette chromosome (SCC)[7–9]. In MRSA, the methicillin-resistant determinant *mecA* is always located within the SCC (SCC*mec*), while its homologs can be found in SCC, chromosomes, or plasmids in *Staphylococcus* (including *S. sciuri* that was recently reclassified to *Mammaliicoccus*[10]), or *Macrococcus* species[11]. SCC*mec* is itself a 20–60-kb genetic element integrated by Ccr (cassette chromosome recombinases) at a specific site (attachment site, *attB*) in *orfX* (a.k.a. *rmlH*, encoding rRNA methyltransferase[12]) near the replication origin of the chromosome[13]. Evolutionary models infer that at least 20 independent acquisitions of SCC*mec* have occurred in *S. aureus*[14]. Although short or fragmented SCC*mec* can be transmitted by transduction[15] or conjugation[16], the exact mechanism of cell-to-cell transmission has been debated for over 50 years[7] (see "Discussion").

The presence of diverse MGEs conveying virulence and resistance factors to the *S. aureus* genome indicates a prominent evolutionary ability mediated by horizontal gene transfer (HGT). Bacteriophage-mediated transduction and conjugative machinery-dependent conjugation are historically well-characterized HGT mechanisms in staphylococci, with the former considered to be the primary method[7]. Another bacterial HGT mechanism, termed natural competence/transformation, refers to the bacterial ability to incorporate extracellular genetic information by expressing competence machinery (DNA incorporation machinery). Competence development is a physiological state that is tightly regulated by certain environmental cues and cellular determinants[17]. The signals governing competence development in Firmicutes are diverse and species dependent, as shown in some model organisms such as *Bacillus subtilis* and *Streptococcus pneumonia*, and are largely unknown in *S. aureus*. In 2012, we reported that a subpopulation of *S. aureus* can develop natural competence for DNA transformation by expressing competence genes under the direct transcriptional control of the cryptic sigma factor SigH[18,19], namely the *comG* operon, encoding the pseudopilus that facilitates DNA access to the channel, and the *comE* operon, encoding an essential DNA internalization channel[17]. Bacteria (N315 derivative strains) modified to overexpress SigH were found to have incorporated SCC*mec*-II elements from purified genomic DNA in a manner dependent on a particular growth medium (CS2)[19]. However, cell-to-cell transfer of SCC has not been demonstrated,

and the transformation efficiencies of the unmodified model strains (N315 or N315ex) and other clinical isolates in that report were under the detection limit ($<10^{-11}$)[19]. The competence transcription factor ComK was also shown to synergistically upregulate many competence genes[20] but efforts to transform test strains by overexpressing SigH and ComK have been unsuccessful[20]. These observations have led to the current belief that natural transformation may not play a major role in staphylococcal evolution, including the multiple, independent emergence of MRSA strains with diverse genetic backgrounds.

Previous analysis showed that *S. aureus* usually does not activate the competence operon promoter ($P_{comG}$) but, when cultivated in CS2 medium, subpopulations expressing $P_{comG}$ reporter increase up to ~10%[19]. This suggests that certain environmental or intrinsic cues are necessary for natural transformation. In the present study, we identify specific two-component systems (TCSs)[21,22] involved in the regulation of the competence operon promoter ($P_{comG}$). Furthermore, we present experimental evidence of inter- and intraspecies transfer of SCC*mec* between staphylococcal cells via natural transformation.

## Results

**TCSs are involved in expression of *comG* in subpopulations.** To delineate conditions conducive to natural transformation, we generated a series of 15 TCS deletion mutants, removing each set of TCS genes (TCS3–TCS17), except the essential TCS1 (WalKR)[23], in *S. aureus* strain N315ex w/o φ[19] (termed Nef) (Table 1). Nef is an N315 derivative strain and does not possess any conjugative elements or lysogenic phage that could transfer DNA by transduction or pseudocompetence. Nef also lacks the SCC*mec* and its embedded TCS2 (*SA0066–SA0067*). The resulting ΔTCS strains in Nef background were designated Δ3–Δ17.

The *comG* operon promoter ($P_{comG}$) has been previously used to monitor SigH-dependent competence gene expression[19]. The promoters of the *comG* and *comE* operons ($P_{comG}$ and $P_{comE}$) are both recognized by SigH[19] and enhanced by ComK[20], suggesting that the regulation of these two operons is under similar control by SigH and ComK. In order to verify whether the $P_{comG}$ reporter is suitable to monitor the expression of competence machinery genes, we introduced the $P_{comG}$-*gfp* and $P_{comE}$-*dsRed* dual-fluorescence reporter plasmid into Nef. As expected, co-expression of these reporters was observed (Supplementary Fig. 1), suggesting that either of the promoters is suitable for monitoring competence gene expression.

Expression of $P_{comG}$-*gfp* was measured in each ΔTCS mutant strain. In planktonic cultures using CS2 medium, the GFP intensity of the parental strain (Nef-GFP) initially increases after 8 h and peaks at around 15 h (Supplementary Fig. 2a). In contrast, no GFP fluorescence could be detected in other media such as tryptic soy broth (TSB) (Supplementary Fig. 2b), in line with our previous observations that activation of the *comG* promoter is dependent upon culture conditions[19].

The ΔTCS strains did not exhibit altered growth curves ($OD_{600}$) in TSB (Supplementary Fig. 3a). However, in CS2 medium, the $OD_{600}$ around 8–12 h for strains Δ5, Δ12, and Δ13 was slightly lowered compared with Nef, while that of strains Δ9 and Δ17 was a bit increased during stationary phase (Supplementary Fig. 3b). Fig. 1a shows the peak values of the reporter expression (intensities of GFP fluorescence per OD) in ΔTCS mutants cultured in CS2 medium. Compared with the parental strain, expression of $P_{comG}$-*gfp* was increased approximately 2.5-fold in the Δ12 strain, significantly lowered (approximately 4-fold) in the Δ13 TCS mutant, and, in strain Δ17, expression was completely abolished, indicating that TCS17 is essential for *comG* expression under these conditions. The remaining ΔTCS

**Table 1 S. aureus N315 has 17 TCSs.**

| | Gene name | Gene locus (SA number in N315) | Function/signal | Ref. |
|---|---|---|---|---|
| TCS1 | walKR | SA0018-SA0017 | Membrane permeability, cell-wall metabolism, autolysis | 21 |
| TCS2 | kdpDE | SA0067-SA0066 | K⁺ transport, virulence-related regulation, stress resistance | 21 |
| TCS3 | hptRS | SA0215-SA0216 | Hexose-phosphate transport | 21 |
| TCS4 | lytSR | SA0250-SA0251 | Autolysis, membrane electrical potential sensor, adaptation to cationic antimicrobial peptides | 21 |
| TCS5 | graSR | SA0615-SA0614 | CAMPs resistance, virulence, stress response, cell-wall signaling, growth at acidic pH | 21,22 |
| TCS6 | saeSR | SA0660-SA0661 | Regulation of exoprotein expression, virulence | 21 |
| TCS7 | | SA1158-SA1159 | Unknown | |
| TCS8 | arlSR | SA1246-SA1248 | Autolysis, cell growth, agglutination, pathogenesis | 21 |
| TCS9 | srrBA | SA1322-SA1323 | Respiratory response, virulence | 21 |
| TCS10 | phoRP | SA1515-SA1516 | Phosphate starvation response (in B. subtillis); Growth during phosphate starvation | 21,68 |
| TCS11 | airSR | SA1667-SA1666 | Oxygen sensing and redox signaling | 21 |
| TCS12 | vraSR | SA1701-SA1700 | Vancomycin resistance; response to cell-wall-targeting antibiotics | 21 |
| TCS13 | agrCA | SA1843-SA1844 | Quorum sensing, virulence | 21 |
| TCS14 | kdpDE | SA1882-SA1883 | K⁺ transport, virulence-related regulation, stress resistance | 21 |
| TCS15 | hssSR | SA2152-SA2151 | Heme sensor system, virulence | 21 |
| TCS16 | nreCB | SA2180-SA2179 | Nitrite and nitrate reduction and transport | 21 |
| TCS17 | braSR | SA2417-SA2418 | Bacitracin and nisin resistance | 21 |

mutations had no significant effect. We also used fluorescence microscopy to observe cell populations expressing the GFP reporter (Fig. 1b), and in the parental strain, 11.3% of the cells expressed GFP after growth for 12–14 h in CS2 medium but, in Δ13 and Δ17, only 2.9% and 0.1% of the cells expressed GFP, respectively (Fig. 1b). In Δ12, on the other hand, 49.3% of the cells expressed GFP, an approximately 4-fold higher expression than the parental strain (Fig. 1b). Complementation in trans for these ΔTCSs restored the parental strain levels of GFP-positive cells, while the vector control had no effect (Fig. 1b). Taken together, our results suggest the involvement of TCS12, TCS13, and, most importantly, TCS17 in the regulation of natural competence in Nef (Fig. 1c).

To test whether the absence of TCS13 and TCS17 can be compensated for by sigH overexpression, we introduced the pRIT-sigHNH7 plasmid that allows constitutive transcription of sigH mRNA with an intact 5′UTR[19] into Δ13 and Δ17 carrying the P$_{comG}$-gfp reporter, generating strains Δ13-NH7-GFP and Δ17-NH7-GFP. Of note, the 5′UTR contains inverted repeat sequences and is thought to suppress SigH translation (Supplementary Fig. 4a). In TSB medium, GFP expression was not observed in Δ13-NH7-GFP, Δ17-GFP-NH7, or Nef-NH7-GFP, indicating that translational suppression is maintained (Supplementary Fig. 4b). In CS2 medium, introduction of the pRIT-sigHNH7 plasmid restored GFP expression in both Δ13 and Δ17, while the vector controls had no effect (Supplementary Fig. 4b). Thus, the absence of TCS13 and TCS17 can be compensated for by the overexpression of sigH mRNA as measured by P$_{comG}$-gfp expression, indicating that these TCSs act upstream from SigH in the competence-regulatory pathway.

**Cell-wall-targeting antibiotics and biofilm-growth conditions modulate comG expression.** We next aimed at identifying environmental conditions that can modulate competence development based on the involvement of TCS12, TCS13, and TCS17. TCS12 (VraSR) is mainly involved in the response and resistance to vancomycin, but also to bacitracin and other antibiotics to some extent[21], while TCS17 (BraSR) is involved in resistance to bacitracin and nisin[24]. Indeed, we confirmed that Δ12 is more susceptible to vancomycin and Δ17 is more susceptible to

bacitracin and nisin than Nef in all tested media (Supplementary Table 1).

In addition, TCS17 has pleiotropic roles in cell physiology, including biofilm formation[25], while TCS13 (AgrCA) is a part of the accessory gene regulator (Agr) quorum sensing system controlling expression of multiple virulence genes[21] by the diffusion-sensing mechanism[26] and is also involved in biofilm regulation. We noted that Δ13 and Δ17 strains were impaired in rigid biofilm formation compared with Nef when cultured statically in CS2 medium (Supplementary Fig. 5a, b); however, this phenotype is not due to impaired growth as colony-forming unit (CFU) counts for Δ13 and Δ17 were not significantly different compared with Nef after 1 day under these conditions (Supplementary Fig. 5b, right). Based on these results, we tested the effects of cell-wall-targeting antibiotics and biofilm-forming, static growth conditions on P$_{comG}$-gfp expression.

Treatment of Nef-GFP with subinhibitory concentrations of vancomycin or bacitracin reduced P$_{comG}$-gfp reporter expression in a concentration-dependent manner (Supplementary Fig. 6a, b). The antibiotics slightly affected cell growth (Supplementary Fig. 6a, b). Thus, the observed change in P$_{comG}$-gfp expression might be due to secondary effects of the antibiotics rather than their direct effects via TCSs. On the other hand, 8 or 16 μg mL$^{-1}$ nisin slightly, but reproducibly, increased reporter expression intensity in Nef-GFP (Supplementary Fig. 6c) though not in a statistically significant manner. No effect of nisin on cell growth was observed (Supplementary Fig. 6c).

To test P$_{comG}$-gfp expression under biofilm-growth conditions, we cultured Nef and its derivative strains statically in CS2 medium, letting cells sediment to the bottom of 6-well polystyrene plates where they stably attach by forming a biofilm (Supplementary Fig. 5b). The sigH-null mutant (ΔH) was used as the negative control. Nef-overexpressing SigH (Nef-H, carrying the pRIT-sigH plasmid with a modified 5′-UTR to release the translational suppression of sigH) was used as a positive control and over 90% of the cells expressed GFP after 1 day (Fig. 2a). In the Nef-GFP reporter strain, the percentage of GFP-expressing cells increased and peaked at day 3 (Fig. 2a), an effect that was impaired in the Δ13 and Δ17 strains. We examined whether other TCSs are involved in P$_{comG}$-gfp expression under these same

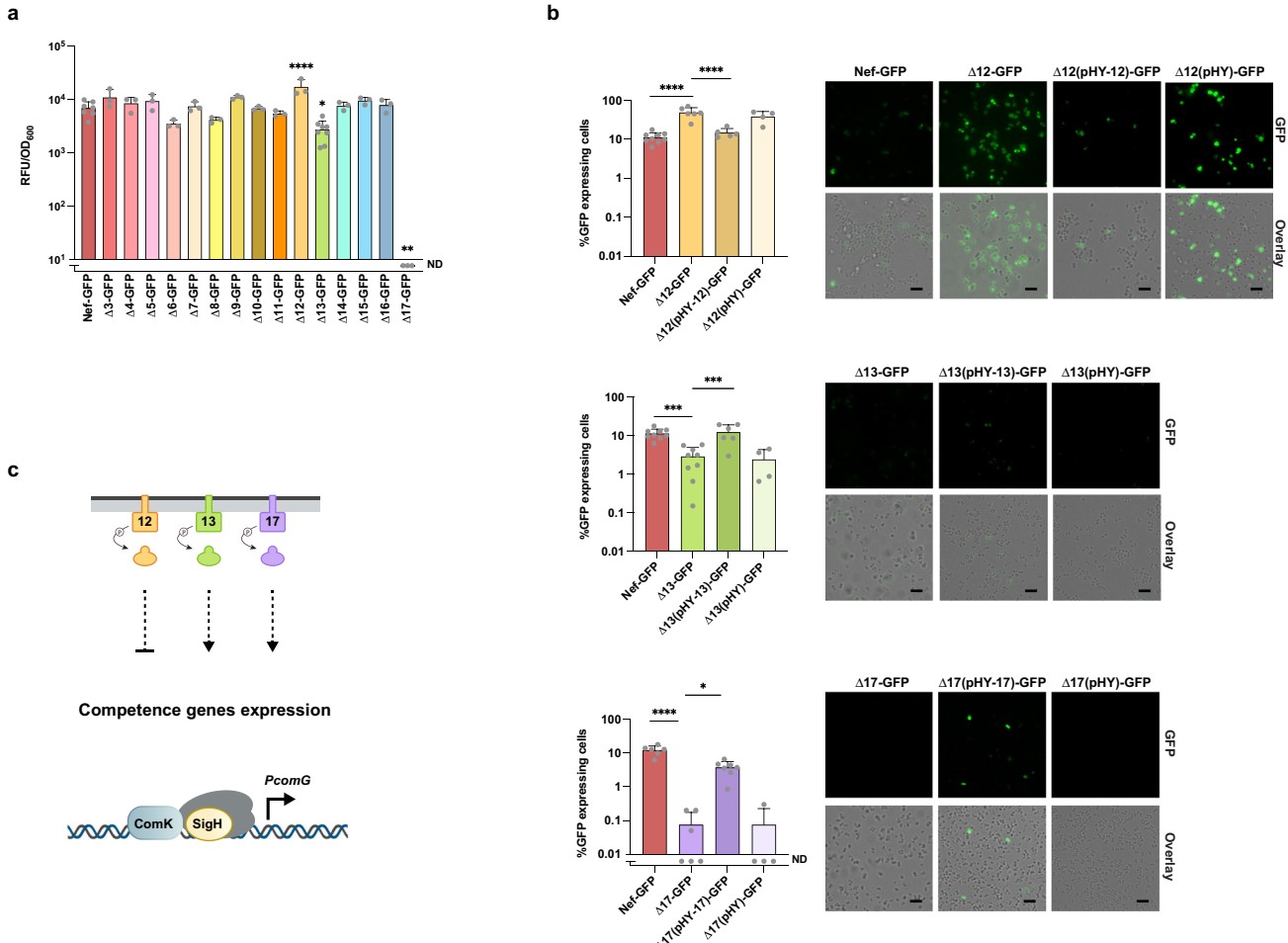

**Fig. 1 TCS12, TCS13, and TCS17 are involved in *comG* promoter activity. a** ΔTCS mutants (Δ3–Δ17) were tested for involvement in the regulation of P*comG*. Nef and its derivative ΔTCS carrying the P*comG*-*gfp* reporter were grown in CS2 medium with shaking. The Y axis shows the increase in RFU/OD$_{600}$ values, which was calculated by subtracting the minimum GFP intensities from the maximum GFP intensities during 12–24 h of growth. Bars represent the mean of at least $n = 3$ independent experiments with error bars indicating SD. Gray dots represent independent experiments. ND: none detected. Statistical significance was determined by one-way ANOVA with Dunnett's multiple-comparison test. $^*P = 0.0127$, $^{**}P = 0.0011$, $^{****}P < 0.0001$. **b** The population percentage expressing GFP was determined after 12–14 h of growth by fluorescent microscopy. TCS mutants were complemented by plasmids (pHY-12, pHY-13, and pHY-17), while the empty vector (pHY) was used as a control. At least 300 cells were counted in each independent experiment. Bars represent the mean of at least $n = 4$ independent experiments with error bars indicating SD. Gray dots represent independent experiments. ND: none detected. Statistical significance was determined by one-way ANOVA with Tukey's multiple-comparison test. $^*P = 0.0348$, $^{***}P = 0.0005$, $^{****}P < 0.0001$. Scale bars, 5 µm. **c** Schematic summary of the TCSs involved in P*comG* regulation. TCS13 (green) and TCS17 (purple) are involved in the activation of P*comG*, while TCS12 (orange) is involved in its suppression. Source data are provided as a Source Data file.

conditions and found that only Δ13 and Δ17 had significantly reduced GFP expression compared with Nef after 3 days (Supplementary Fig. 7), whereas Δ7, Δ9, and Δ12 had significantly increased GFP expression. Complementation in Δ13 and Δ17 restored P*comG*-*gfp* expression to levels equivalent to Nef values in the biofilm (Fig. 2b).

In conclusion, these data indicate that P*comG*-*gfp* expression is affected by cell-wall-targeting antibiotics as well as environmental cues or cellular status in biofilm growth where TCS13 and TCS17 play important roles.

**Biofilm-growth conditions enhance natural transformation efficiency**. We next tested the effects of nisin and biofilm conditions on transformation efficiency. In the planktonic transformation assay described previously[19], nisin (8 µg mL$^{-1}$) had no detectable effect on transformation; Nef remained nontransformable, irrespective of nisin, while the transformation

frequency of Nef-H was not affected by nisin (Supplementary Fig. 8).

To detect transformants in cells grown under biofilm conditions, it was necessary to use heat-killed donor cells rather than purified DNA (see "Discussion"). Strains carrying a chromosomal tetracycline- resistance gene were used as a donor (N315Δcls2-tet$^R$, or NefΔcls2-tet$^R$). Under biofilm-forming conditions, the transformation frequency in Nef reached 10$^{-6~7}$ at day 3 (Fig. 2c, left), which remained undetectable in planktonic growth conditions (undetected, <10$^{-11}$, $n = 5$) (Fig. 2c, right). In Nef-H, the transformation frequency was similar to Nef under biofilm-forming conditions (Fig. 2c, left), while ΔH was nontransformable, indicating that SigH expression is essential but not the limiting factor for transformation in biofilm. We confirmed that no spontaneous tetracycline-resistant colonies emerged in our experimental conditions, and that the transformants share the same genetic backbone as the recipient (Supplementary Fig. 9)[27].

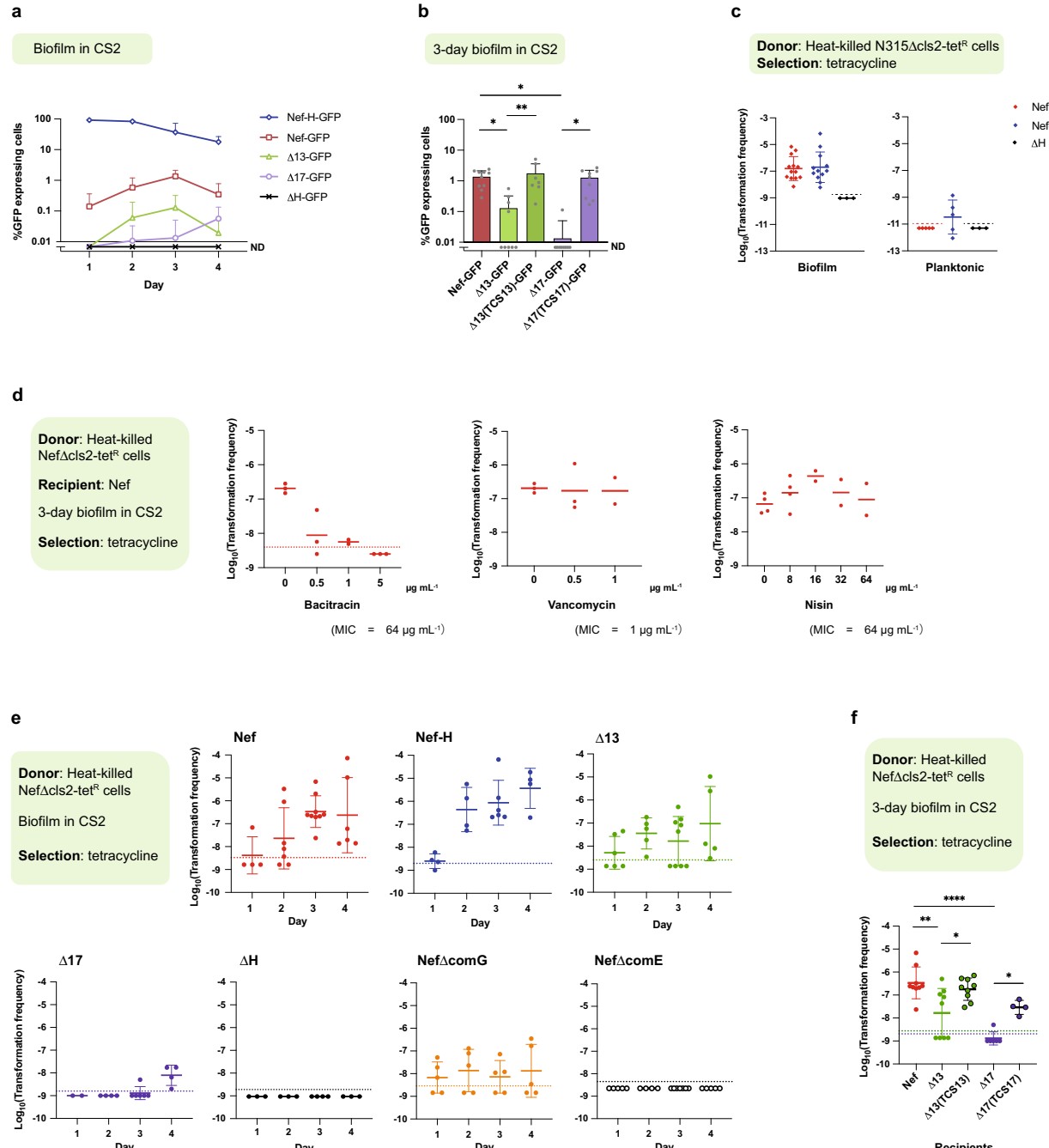

**Fig. 2 TCSs 13 and 17 are important for *comG* promoter activity and natural transformation in CS2 static (biofilm) growth conditions. a** The percentage of cells expressing P*comG*-*gfp* reporter in Nef and its derivatives, Nef-H (overexpressing SigH), Δ13, Δ17, and ΔH (*sigH*-null mutant). Cells were statically grown in CS2 medium. The mean of at least *n* = 3 independent experiments is shown with error bars indicating SD. ND: none detected. **b** Chromosomal complementation of the mutants restores the percentage of cells expressing GFP at day 3 in the biofilm-growth conditions. Bars represent the mean of at least *n* = 7 independent experiments with error bars indicating SD. Gray dots represent independent experiments. ND: none detected. Chromosomally complemented strains were used for consistency with the transformation experiments (Fig. 2f). Statistical significance was determined by one-way ANOVA with Tukey's multiple-comparison test. Nef-GFP vs. Δ13-GFP *P* = 0.0439, Nef-GFP vs. Δ17-GFP *P* = 0.0223, **P* = 0.0093, Δ17-GFP vs. Δ17(TCS17)-GFP *P* = 0.0466. **c–f** Transformation assays. Data points represent independent experiments. The dotted lines represent detection limits. The mean of at least *n* = 2 independent experiments is shown. Error bars indicate SD. **c** Transformation frequencies of Nef (red), Nef-H (blue), and ΔH (black) at day 3 in the planktonic or biofilm-growth conditions. Transformants were selected by tetracycline. **d** Nef was statically grown with bacitracin, vancomycin, or nisin in CS2 medium. Transformants were selected by tetracycline after 3 days. **e** Time-course development of natural transformation in the biofilm for 4 days. Nef and its derivatives, Nef-H, Δ13, Δ17, ΔH, NefΔcomG (*comG* operon-null mutant), and NefΔcomE (*comE* operon-null mutant) were statically grown in CS2 medium. Natural transformation frequencies were determined every 24 h. **f** Complementation of the TCS13 and TCS17 restores the transformation frequencies at day 3 in the biofilm. Chromosomally complemented strains were used because plasmid-based complemented strains are tetracycline resistant. Statistical significance was determined by one-way ANOVA with Tukey's multiple-comparison test. Δ13 vs. Δ13(TCS13) *P* = 0.0272, Δ17 vs. Δ17(TCS17) *P* = 0.0386, **P* = 0.0031, ****P* < 0.0001. Source data are provided as a Source Data file.

We tested the effects of bacitracin, vancomycin, or nisin on the transformation efficiencies of Nef cells growing in biofilms (Fig. 2d). Low-concentration bacitracin treatment ($0.5 \, \mu g \, mL^{-1}$) reduced the transformation efficiency, while $5 \, \mu g \, mL^{-1}$ completely prevented the detection of transformants in Nef (Fig. 2d, left). Bacitracin also reduced biofilm formation (Supplementary Fig. 5c). Vancomycin treatment, on the other hand, had no significant effect on natural transformation in this strain at all tested concentrations (Fig. 2d, middle). Nisin treatment at $16 \, \mu g \, mL^{-1}$ increased the transformation efficiency (Fig. 2d, right). The effect of cell-surface affecting antibiotics is pleiotropic and, while their effects on $P_{comG}$ activity/transformation might be indirect, our data indicate that they may offer some degree of control over natural transformation efficiency.

Figure 2e shows the time course for the transformation frequencies of Nef, Nef-H, Δ13, Δ17, ΔH, NefΔcomG, and NefΔcomE. Transformation efficiencies of Nef increased up to day 3. No transformants could be detected in ΔH and NefΔcomE, confirming that transformation under biofilm conditions is entirely dependent on SigH and the essential DNA channel encoded by the *comE* operon. Deletion of the *comG* operon (NefΔcomG) severely impaired transformation in biofilms but did not abolish it completely. The nonessentiality of the *comG* operon encoding the pseudopilus is consistent with reports in model species[17]. In *Bacillus* spp., it is also known that the ComG pseudopilus is dispensable for DNA binding by ComEA in the absence of a cell wall[28]. Thus, under biofilm conditions where DNA availability and cell-to-cell contact are increased, it is conceivable that the requirement for the ComG-encoded pseudopilus is reduced.

Transformation frequencies of Δ13 and Δ17 were significantly reduced at day 3 compared with Nef (Fig. 2f) but complementation restored them (Fig. 2f). In addition, overexpression of either *sigH* mRNA by pRIT-sigHNH7 or SigH protein by pRIT-sigH restored biofilm formation and transformation frequencies in Δ13 and Δ17 (Supplementary Fig. 10).

For experiments under planktonic growth conditions, CS2 medium was indispensable for detecting transformation[19,27]. We therefore tested whether static biofilm conditions could induce transformation with other growth media such as TSB, brain–heart infusion (BHI), RPMI, M9 supplemented with amino acids (M9 + aa), or MB (whole milk and BHI [1:1] ratio) (Supplementary Fig. 11). Transformation was detected in Nef in all growth media tested with diverse frequencies and was the highest in CS2 and MB. NefΔcomE was nontransformable in any media (Supplementary Fig. 11a). Biofilm formation was variable, dependent upon culture media, and was most efficient in CS2 and MB (Supplementary Fig. 11b).

To gain further insight into how biofilm formation and transformation are related to TCS13 and TCS17 functions, we tested transformation of Δ13 and Δ17 during growth in MB medium where we found that both mutants could make rigid biofilms (Supplementary Fig. 12a). The transformation frequencies of Δ13 and Δ17 were comparable to Nef in MB (Supplementary Fig. 12b), suggesting that TCS13 and TCS17 indirectly contribute to transformation via biofilm formation in CS2 but are dispensable in MB.

**Clinical isolates are capable of natural transformation in biofilm.** As natural transformation in *S. aureus* has only been detected in N315 derivative strains that were genetically engineered to overexpress SigH[19], we tested the transformability of 5 unmodified clinical isolates (tetracycline sensitive) during growth under CS2 biofilm conditions. We found that one strain (MRSA, r59) was transformable by the chromosomal tetracycline-resistance marker

(Supplementary Fig. 13a). $P_{comG}$-*gfp* reporter analysis indicated that strains r59 and s142 express GFP at frequencies of ~1% and ~0.1%, respectively, while s1587, r3, and r408 did not (Supplementary Fig.13b). Strains s142 and r3 became transformable after introducing a SigH-expressing plasmid (pRIT-sigH), suggesting that SigH expression remains a limiting step in transformation of s142 and r3 (Supplementary Fig. 13a). Two strains (s1567 and r408) were not transformable, irrespective of the SigH-expressing plasmid. When Nef (tetracycline-sensitive negative control) was used as the donor, no colonies emerged in any of the recipients (Supplementary Fig. 13c). Taken together, these data indicate that biofilm-growth conditions facilitate transformation in some, but not all, clinical isolates.

**SCCmecA elements can be transferred by natural transformation in biofilms.** Exploring the SCC transfer mechanism is challenging, as small SCC elements can be transferred by transduction[15], but typical staphylococcal-transducing bacteriophages are *Siphoviridae* with genome sizes of less than ca. 45 kb[29] and cannot physically accommodate an entire large SCC. Conjugation can also transfer a shortened SCCmec II but this requires insertion of SCC into the conjugative plasmid[16] and, to the best of our knowledge, *mecA* has not been found in such a plasmid-carrying SCC in staphylococcal isolates. In this study, we propose that natural transformation in biofilms mediates cell-to-cell transfer of SCCmecA based on the following experimental evidence.

We tested transfer of *mecA* by natural transformation in biofilms using MSSA clinical isolates as recipients and heat-killed MRSA or methicillin-resistant coagulase-negative staphylococci strains (MR-CoNS) as donors. The *mecA* transformants were selected by cefmetazole (from the cephem subgroup of the β-lactam antibiotics) and their ability to grow in the presence of cefmetazole was confirmed by replica method (Supplementary Fig. 14). We found that approximately 20 strains out of 99 various clonal complex (CC) types were able to form colonies under cefmetazole selection (Supplementary Table 2). Among these, 5 (1s, 9s, 11s, 35s, 98s) were selected for further analysis. These 5 MSSA strains, together with Nef and NefΔcomE as positive and negative controls, were tested for their transformability with distinct staphylococcal species and SCC types (Fig. 3: transformation frequency, Supplementary Table 3: number of transformants). *S. aureus* or MR-CoNS, along with any tested SCCmec (I, II, III, IVa), served as donors with detected efficiencies ranging from ca. $10^{-8}$ to $10^{-7}$, generating up to ~ 160 colonies from a single-well biofilm containing $10^9$ CFU recipient cells (Supplementary Table 3). We deleted the *comE* operon of 9s and found that this mutant was nontransformable using N315 as the donor (Fig. 3a). No transformants could be detected when Nef (carrying no SCCmec) was used as the donor (Fig. 3a).

These MSSA recipients were transformable by the chromosomal tetracycline-resistance gene. while no transformation was detected using the pT181 plasmid in any transformable strain (Fig. 3b). The question as to transfer of other plasmids by natural transformation under biofilm conditions remains unanswered.

**Confirmation of SCCmec acquisition and stability in transformants.** Figure 4a shows the result of *mecA* PCR for multiple transformants. All transformants derived from 35s showed the *mecA* signal and the minimum inhibitory concentrations (MICs) of cephems (cefmetazole and cefoxitin) were increased in these transformants, demonstrating their conversion to MRSA (Supplementary Table 4). In contrast, some transformants showed lower intensities in *mecA* signal compared with 35s transformants (Fig. 4a). Moreover, the MIC values in these transformants were

**a**

3-day biofilm in CS2
**Selection**: cefmetazole

| Heat-killed donor | | SCC type | Nef | NefΔcomE | NefattB* | 1s | 9s | 9sΔcomE | 11s | 35s | 98s |
|---|---|---|---|---|---|---|---|---|---|---|---|
| *S. aureus* | COL | I | $3.3\times10^{-8}$ (n = 1), ND (n = 1) | ND (n = 2) | | $1.4\times10^{-8}$ (n = 2) | $2.4\times10^{-8}$ (n = 2) | | $9.1\times10^{-8}$ (n = 2) | $1.7\times10^{-8}$ (n = 2) | |
| | COLw/oφ | I | $1.2\times10^{-8}$ (n = 2) | ND (n = 3) | | $2.2\times10^{-8}$ (n = 2) | $1.2\times10^{-8}$ (n = 2) | | $1.6\times10^{-8}$ (n = 2) | $2\times10^{-8}$ (n = 2) | $2.5\times10^{-8}$ ± $1.8\times10^{-8}$ (n = 3) |
| | N315 | II | $7.2\times10^{-8}$ ± $6.8\times10^{-8}$ (n = 5) | ND (n = 4) | ND (n = 6) | $3.7\times10^{-8}$ ± $4.4\times10^{-8}$ (n = 5) | $9.9\times10^{-9}$ ± $4.2\times10^{-9}$ (n = 6), ND (n = 2) | ND (n = 4) | $2.0\times10^{-8}$ ± $1.1\times10^{-8}$ (n = 5) | $4.9\times10^{-8}$ ± $4.1\times10^{-8}$ (n = 7) | |
| | N315ΔccrAB | II (ΔccrAB) | ND (n = 4) | ND (n = 6) | ND (n = 3) | | ND (n = 3) | ND (n = 3) | | | |
| | Nef | none | | ND (n = 2) | ND (n = 4) | ND (n = 3) | ND (n = 3) | | ND (n = 3) | ND (n = 3) | |
| | 35s[CoNS17] | IVa | $9.3\times10^{-9}$ (n = 1), ND (n = 1) | ND (n = 2) | | $5.6\times10^{-8}$ (n = 2) | $1.4\times10^{-8}$ (n = 2) | | $1.7\times10^{-9}$ (n = 1), ND (n = 1) | $1.1\times10^{-8}$ (n = 2) | |
| | MW2 | IVa | $2.1\times10^{-8}$ (n = 2) | ND (n = 2) | | $4.9\times10^{-8}$ (n = 2) | $5.2\times10^{-8}$ (n = 2) | | $3.3\times10^{-8}$ (n = 2) | $3.7\times10^{-8}$ (n = 2) | |
| MR-CoNS | CoNS16 | I | $9.6\times10^{-9}$ (n = 1), ND (n = 1) | ND (n = 5) | ND (n = 3) | $2.1\times10^{-8}$ (n = 2) | $5.0\times10^{-8}$ (n = 1), ND (n = 1) | | $2.5\times10^{-8}$ (n = 2) | $5.4\times10^{-8}$ ± $7.3\times10^{-8}$ (n = 3) | |
| | CoNS9 | III | $6.2\times10^{-8}$ (n = 2) | ND (n = 2) | | $9.7\times10^{-8}$ (n = 1) | $3.5\times10^{-8}$ ± $1.4\times10^{-8}$ (n = 3) | | $5.4\times10^{-8}$ (n = 1) | $3.3\times10^{-8}$ (n = 2) | |
| | CoNS10 | IVa | $1.9\times10^{-7}$ (n = 2) | ND (n = 5) | ND (n = 3) | $5.5\times10^{-8}$ (n = 1) | $2.7\times10^{-8}$ (n = 1) | | $3.4\times10^{-8}$ ± $1.9\times10^{-8}$ (n = 3) | $2.9\times10^{-8}$ (n = 2) | |
| | CoNS11 | IVa | $1.2\times10^{-7}$ ± $1.6\times10^{-7}$ (n = 3) | ND (n = 5) | ND (n = 4) | $8\times10^{-8}$ (n = 2) | $2.2\times10^{-8}$ ± $1.2\times10^{-8}$ (n = 3) | ND (n = 2) | $4.2\times10^{-8}$ (n = 2) | $5.1\times10^{-8}$ (n = 2) | |
| | CoNS15 | IVa | $1.2\times10^{-8}$ (n = 2) | ND (n = 5) | ND (n = 3) | $5.25\times10^{-8}$ (n = 2) | $1.9\times10^{-8}$ (n = 2) | | $1.8\times10^{-8}$ (n = 2) | $3.4\times10^{-8}$ (n = 2) | |
| | CoNS17 | IVa | $2.1\times10^{-8}$ (n = 2) | ND (n = 5) | ND (n = 3) | $4.1\times10^{-8}$ ± $3.3\times10^{-8}$ (n = 5) | $2.6\times10^{-8}$ ± $1.2\times10^{-8}$ (n = 4) | | $2.6\times10^{-8}$ (n = 1) | $2.2\times10^{-8}$ (n = 2) | |

**b**

3-day biofilm in CS2
**Selection**: tetracycline

| Heat-killed donor (Tet$^R$) | Nef | NefΔcomE | NefattB* | 1s | 9s | 11s | 35s | 98s |
|---|---|---|---|---|---|---|---|---|
| NefΔcls2-tet$^R$ | $2.5\times10^{-7}$ ± $2.9\times10^{-7}$ (n = 4) | ND (n = 2) | $5.8\times10^{-8}$ (n = 2) | $2.7\times10^{-9}$ (n = 2), ND (n = 2) | $1.3\times10^{-8}$ (n = 2), ND (n = 2) | $1.0\times10^{-8}$ ± $1.2\times10^{-8}$ (n = 3), ND (n = 1) | $2.9\times10^{-8}$ (n = 1), ND (n = 1) | $7.1\times10^{-9}$ (n = 2) |
| Nef-pT181 | ND (n = 2) | ND (n = 2) | | ND (n = 2) | ND (n = 2) | ND (n = 2) | ND (n = 2) | ND (n = 2) |

**Fig. 3 Intra- and interspecies transformation of distinct SCC*mec* elements in biofilm-growth conditions. a**, **b** Transformation frequencies were determined at day 3 in the CS2 biofilm conditions. The mean of frequencies (number of transformants/CFU of recipient) is shown ±SD. n: number of independent experiments. ND: none detected (c.a. <10$^{-9}$). **a** The *mecA* transformation was tested using MRSA or MR-CoNS donors and MSSA recipients. Transformants were selected by cefmetazole. Nef has no *mecA* and served as a negative-control donor. NefΔcomE is the nontransformable negative-control recipient. **b** Transformability of each recipient was tested by using donors carrying the tetracycline-resistance gene. NefΔcls2-tet$^R$ carries *tet* in the chromosome, while Nef-pT181 carries *tet* on the plasmid. Source data are provided as a Source Data file.

relatively lower than 35s transformants, suggesting that 35s, but not others, could stably accommodate the *mecA* gene. Propidium monoazide (PMA)–PCR and DNase treatment excluded the possibility that the *mecA* signal was from contaminating extracellular donor DNA as it could still be detected in the stable 35s[N315] and unstable 11s[N315] transformants (Supplementary Fig. 15). The stability test of cefoxitin resistance showed that the 35s and 98s transformants sustained resistance in the absence of β-lactam, but 1s, 9s, and 11s derivatives tested swiftly lost their resistance (Fig. 4b). Disk-diffusion tests (Fig. 4c) confirmed the reduced susceptibility of the stable transformant 35s[CoNS15] to cefoxitin (CFX) and oxacillin (MPI). On the other hand, the unstable 9s[CoNS15] was categorized as MSSA even though the CFX inhibitory zone slightly decreased and colonies appeared on the edge of the MPI inhibitory zone. Notably, we detected stable 9s transformants when N315 was used as donor (Fig. 4b). This suggests that SCC*mec* types and recipient strains are drivers of SCC*mec* stability in transformants. Stability and SCC intactness in transformants are summarized in Fig. 4d.

PCR analysis showed that the full-length transferred SCC*mec* IVa was present in the stable 35s transformants (Fig. 4e) and SCC*mec* I in the stable 98s transformants (Fig. 4f). Full-size SCC*mec* II was detected in transformants of Nef and 9s but was shortened reproducibly in 35s transformants due to the elimination of mobile elements Kdp, Tn554, and IS431-pUB110 (Fig. 4g). Integration of SCC*mec* into the recipient chromosomes of 9s and 35s was confirmed by PCR (Supplementary Fig. 16) and genome sequencing analysis (Supplementary Fig.17).

Mannitol utilization and multiplex PCR analysis[30] also confirmed that obtained transformants are MRSA and had the same genetic backbone as recipient strains (Supplementary Fig. 18).

Collectively, these observations demonstrate that SCC*mec* elements can be transferred to MSSA strains by natural transformation in biofilms.

**SCC*mec* transformation depends on *ccrAB* recombinase genes and an intact *attB* site.** SCC carries *ccr* genes encoding a dedicated excision and integration system but there is scarce evidence

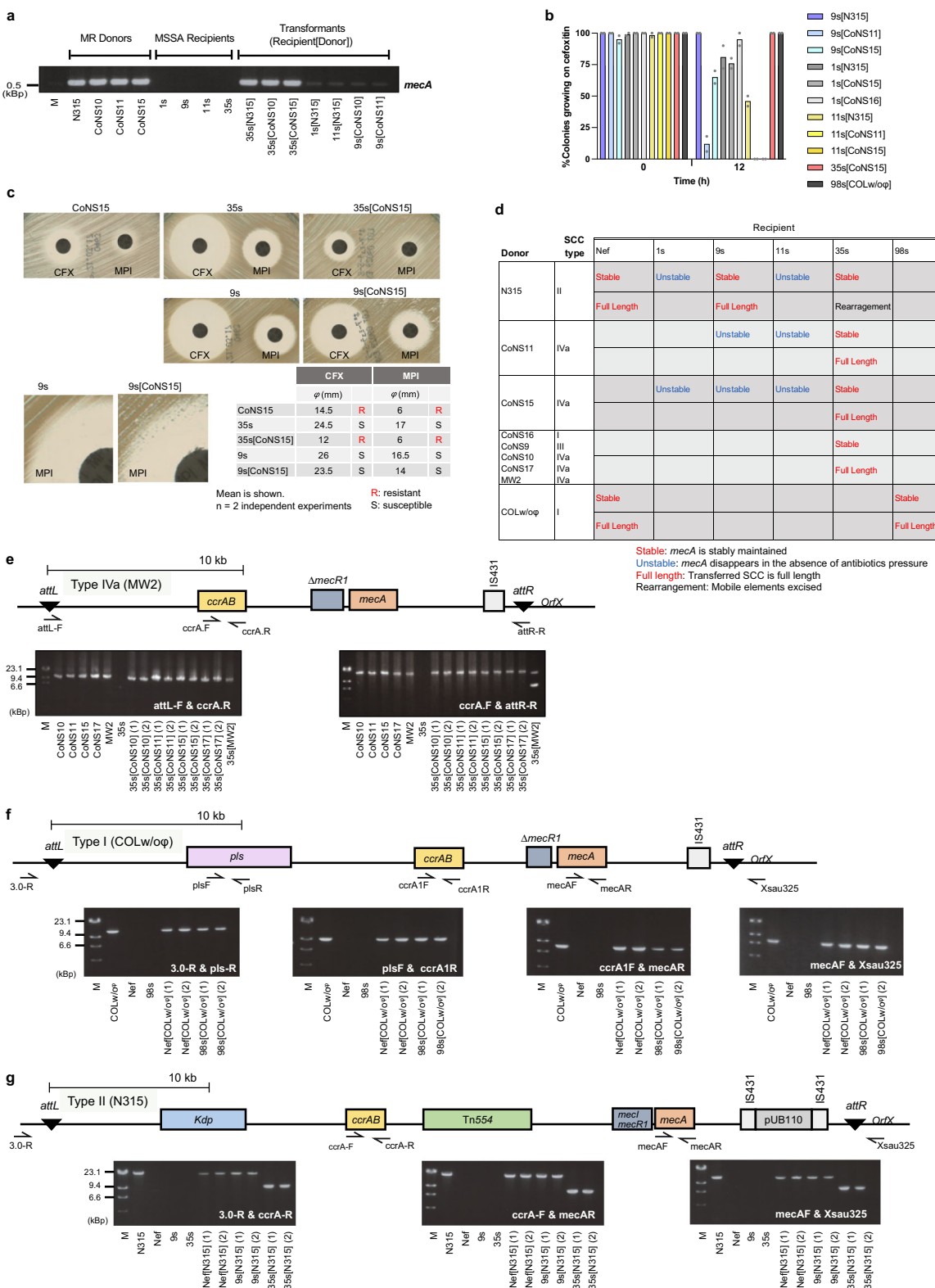

of the mechanistic requirements of this system for intercellular HGT. To determine whether SCCmec transformation is mediated by CcrAB (cassette chromosome recombinases), we deleted the ccrAB gene from the SCCmec-II element (N315ΔccrAB) and evaluated its ability to serve as SCCmec donor for 9s and Nef strains. SCCmec transformants could not be obtained using N315ΔccrAB as a donor while its parental strain, N315, could serve as the SCCmec donor (Fig. 3a). This suggests that the site-

specific excision/integration of SCCmec mediated by ccrAB is essential for transformant generation. We also generated mutations in the attB sequence on the recipient side (NefattB*). This mutant did not generate SCCmec transformants but was transformable with a chromosomal tetracycline-resistance marker (Fig. 3). This evidence points to the ccrAB–attB-dependent SCC transfer system as critical for intercellular SCCmec transformation.

**Fig. 4 SCC*mec* transformation. a** Presence of the *mecA* gene was verified by PCR using mecAF and mecAR primers with equal amounts of DNA template. DNA of donor and recipients was used for positive and negative controls. M: DNA size marker, λ HindIII. The experiment was repeated at least twice independently with similar results. **b** Stability of resistance. The transformants were passaged in drug-free media for 12 h after growth with cefoxitin (4 μg mL$^{-1}$) for 12h (Time 0). The percentage of cells that can grow on cefoxitin was calculated by the replica method. Bars represent the mean of $n = 2$ independent experiments. Data points representing independent experiments are shown by gray dots. **c** Disk-diffusion test of β-lactam antibiotics. CFX: cefoxitin, MPI: oxacillin. Rght bottom: diameters of inhibitory zones. **d** Intactness and stability of SCC*mec* in transformants. **e–g** Schematic structures of SCC*mec* IVa in MW2 chromosomal DNA (**e**), SCC*mec* I in COLw/oφ (COL without phage) chromosomal DNA (**f**), and of SCC*mec* II in N315 chromosomal DNA (**g**). Primer locations are indicated with arrows. Primers attL-F and attR-R can also amplify the nonintegrated SCC*mec* IVa (**e**). Primers 3.0-R and Xsau325 locate outside of SCC and specifically amplify the chromosomally integrated SCC*mec* I and II (**f, g**). DNA of MR donors and MSSA recipients was used for positive and negative controls. Suffix (1), (2) represents transformants obtained from two independent experiments. (**e–g**, bottom) Long PCR verification of the entire SCC*mec* IVa (**e**), I (**f**), or II (**g**) elements in transformants where *mecA* was present. M: DNA size marker, λ HindIII. Source data are provided as a Source Data file.

## Discussion

This study demonstrated intercellular transmission of SCC*mec* by natural transformation and provides mechanistic information on the pathway of MRSA emergence. SCC, an MGE shared among staphylococcal species and *Macrococcus caseolyticus*[31], is responsible for the dissemination of virulence factors and resistance genes such as capsule-synthesis genes (SCC*cap*), the fusidic acid- resistance gene (SCC*fus*), and the methicillin-resistance gene (SCC*mec*) (see comprehensive review[11]). Since its discovery, SCC*mec* has been the focus of extensive research efforts to clarify the global emergence and dissemination of MRSA. At least 13 different SCC*mec* types (I–XIII) have been described to date[11].

Ccr recombinases were found to mediate the excision and insertion of SCC at the *attB* locus (*attL*/*attR* after SCC*mec* integration)[13], with CcrA and CcrB for SCC types I–IV and CcrC for type V[32]. These *ccrAB* genes are expressed in minor subpopulations and the excised circular SCC is thought to serve as a donor for horizontal transmission[33]. Both proteins are required for the proper excision of SCC*mec* from the chromosome and its integration into the *attB* site after transduction as a part of the artificial plasmid[34].

Despite well-established epidemiological evidence of interspecies SCC*mec* movement and *ccr*-dependent excision/insertion, the major intercellular transmission mechanism has remained undefined for half a century. Transfer of the methicillin-resistance gene was first demonstrated by transduction[35] and by pseudocompetence[36] which, at that time, was described as "transformation" but, after discovery of the phage component, is now termed "pseudo-competence" or "pseudo-transformation". Transduction has been suggested as a preferable transfer route for some types of SCC*mec* (albeit with a <45-kb capacity limit of the bacteriophage capsid) and SCC*mec* fragments are detectable in bacteriophage capsids[37,38], making the transfer of short SCC*mec* (types IVa and I) observable by transduction among compatible strains[15]. However, major deletions are occasionally associated with transduction[15] and successful integration into the recipient chromosome requires homologous flanking sequences, suggesting that this transduction relies on homologous recombination rather than the *ccr*-mediated system. Conjugation has also been suggested as a possible mechanism for SCC*mec* transfer[16]. However, successful transfer requires donor manipulation by overexpressing the *ccr* recombinase to capture a shortened SCC*mec* into a conjugative plasmid[16] while spontaneous and large element transfers have not been demonstrated. Nonetheless, the finding of *mecB*-carrying plasmid with conjugative elements in *S. aureus* suggests a possible role of conjugation in transferring SCC-carrying plasmids[39,40].

The SCC transformation observed in this study was dependent on CcrAB-mediated excision/integration and, based on this evidence, we propose natural transformation as a route for SCC*mec* transmission. The observed SCC transformation efficiency of up

to 10$^{-7}$ (CS2: Fig. 3, MB: Supplementary Fig. 12c), plus the fact that SCC transformation is observed in multiple genetic backgrounds, could explain the historical, independent transfers of distinct SCC types to *S. aureus*. Transduction remains as a candidate HGT mechanism for short SCC*mec,* but the extent of *ccr* involvement in this process remains elusive.

In this study, we used heat-killed donors unable to synthesize conjugative machinery, eliminating the possibility of conjugation. Regarding transduction, heat may not completely destroy the infectious phage particles in donors and many of the tested donor strains had phages (Supplementary Fig. 19a). However, the finding that COLw/oφ (phageless) can serve as SCC*mec* donor (Fig. 3a) indicates independence from phage-dependent systems such as transduction. Importantly, the combination of COLw/oφ donor and Nef recipient resulted in no phages in the experimental system. We would like to also emphasize that, at least in the combination of heat-killed COLw/oφ (as SCC*mec* donor) with Nef or 9s recipients, the *comE* operon genes were essential for SCC*mec* transfer, clearly demonstrating that SCC*mec* was transferred by natural genetic transformation rather than phage-dependent systems or conjugation.

It has been suggested that *mecA* acquisition and expression in *S. aureus* has a fitness cost and the process of obtaining β-lactam resistance is complex, involving multiple mutations and metabolic adaptations[41–43]. It was previously suggested that different *S. aureus* genetic backgrounds offer different capacities to accommodate *mecA*[44] and we observed a similar effect as Nef and three of our tested MSSA strains (9s, 35s, and 98s) could stably accommodate the transferred *mecA*, whereas 1s and 11s could not (Fig. 4d). In line with a report that β-lactamase regulators affect *mecA* regulation[45], the tested MSSA strains that could be transformed with SCC*mec* (1s, 9s, 11s, 35s) had plasmids and, importantly, were positive in *blaZ* and *blaI* genes usually localized to plasmids (Supplementary Table 2). The sequences encompassing the *attB* site are also known to affect the efficiency of chromosomal SCC integration[46] and our tested MSSA strains were diverse in the *attB* downstream sequences (Supplementary Fig. 20). Although the present study did not address the factors that stabilize the SCC*mec*, the methodology of SCC*mec* transfer established here would be invaluable to detail the factors that define *mecA* stability, including involvement of DNA methylation, restriction-modification systems, or CRISPR.

The present study establishes a reliable experimental system to detect natural transformation in *S. aureus*. Based on our results, a regulatory framework for competence development is shown in Fig. 5. All tested factors that positively affect biofilm formation increased transformation efficiency, such as static growth in CS2 (Fig. 2) and TCS13 or TCS17 in CS2 (Fig. 2, Supplementary Fig. 5). Notably, MB medium that facilitates biofilm formation allowed *tet* and *mecA* transformation in TCS13 and TCS17 mutants (Supplementary Fig. 12b, c). This indicates that these signaling systems are

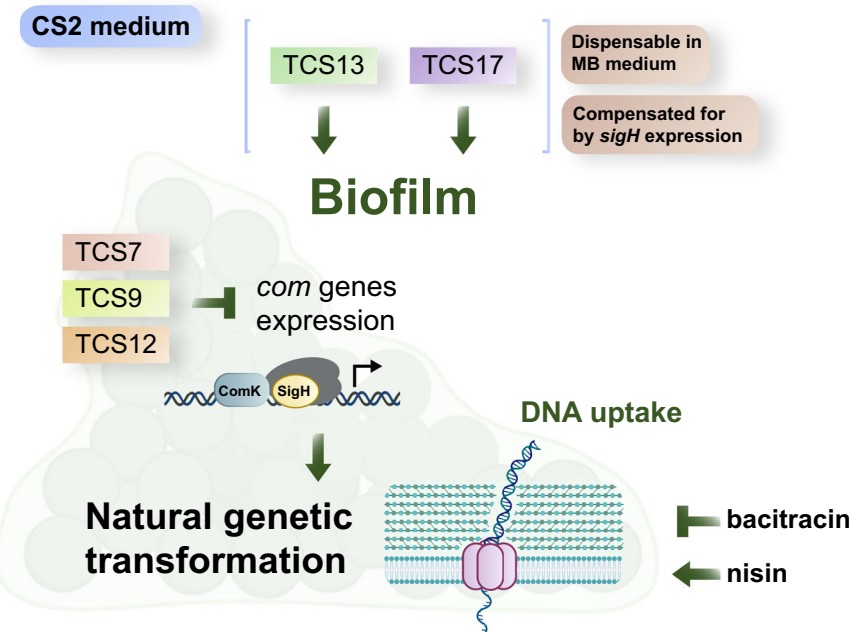

**Fig. 5 Factors affecting competence development in biofilms.** In CS2 medium, TCS13 and TCS17 are necessary for biofilm formation and transformation but can be compensated for by *sigH* expression. In MB medium biofilm, TCS13 and TCS17 are not mandatory for transformation. In CS2 medium, TCS7, TCS9, and TCS12 are not required for biofilm formation but do affect the expression of *com* genes. Biofilms may also facilitate transformation due to better or more sustained cellular contact between donor and recipient cells, reducing the necessity of the ComG pseudopilus, while ComE DNA internalization channels remain essential. Bacitracin and nisin affect natural transformation in biofilms, though detailed mechanisms remain elusive. The figure was partially created with BioRender.com.

dispensable once biofilm forms; however, how the biofilm structure or matrix components increase transformation efficiency remains to be determined. The expression of $P_{comG}$ in CS2 biofilm was under the control of multiple TCSs, including TCS9 (SrrBA), 12, 13, and 17 (Supplementary Fig. 7), in addition to SigH and ComK. SrrBA is a redox-responsive transcription-regulatory system that also includes AgrA (response regulator of TCS13)[47], indicating that microanaerobic redox status in biofilms might affect transformability. Supporting this idea, we previously reported that $P_{comG}$ reporter expression is increased under anaerobic conditions in certain synthetic media such as CS1 or -GS[19].

In our previous study, purified genomic DNA could serve as the donor of SCC in the transformation of planktonic SigH-expressing cells. In contrast, in order to detect SCC transformation in biofilm, it was crucial to use heat-killed donor cells (cellular DNA resides within a peptidoglycan cell wall that cannot be disrupted by heat) rather than purified chromosomal DNA. This might be consistent with the fact that nuclease production is a common characteristic across all strains of *S. aureus* and also occurs in biofilms[48]. We confirmed that DNase-I treatment (e.g., $10\,\mu g\,mL^{-1}$, less than in Supplementary Fig. 15) cannot completely degrade DNA in heat-killed cells. Alternatively, it is possible that experimentally added, purified DNA cannot serve as a transformation donor since extracellular DNA is known to be tightly sequestered in biofilm[49,50]. It also remains to be established whether certain donor factors facilitate transformation or if certain recipient activities (e.g., peptidoglycan hydrolases) cooperatively work to retrieve the DNA packed and protected in the cell wall. It is interesting to note that, in *Streptococcus pneumoniae*, noncompetent cells undergo lysis by bacteriocins and fratricins released by neighboring competent cells[51] but the presence of such a dedicated mechanism for DNA supply is not known in the *Staphylococcus* genus.

Staphylococcal infections are intimately associated with biofilm formation[52] which provides protection against antimicrobial treatment and host-clearance mechanisms[53] while contributing to the prolonged infection and colonization that facilitates the dissemination of drug-resistant strains[54]. Our finding that *S. aureus* can develop natural transformation in biofilm conditions emphasizes the additional role of biofilms in promoting HGT as well as transduction and conjugation[7,55]. In addition, cells release phages at higher frequencies than under planktonic growth conditions[56] and subsequent cell lysis in biofilms would create an ample supply of genetic material for nonlysed cells. Interestingly, expressing SigH was shown to stabilize phage lysogeny[57], implicating a co-evolution of distinct HGT mechanisms in staphylococcal biofilms. Mixed biofilms of *S. aureus* and other staphylococci are thus general hotspots for HGT.

One limitation of this study is that we did not examine natural conditions for MRSA emergence. It is unknown whether this occurs in some host organisms or certain abiotic environments. Further study is necessary but polymicrobial biofilms in which CoNS and *S. aureus* coexist would be likely places to observe this phenomenon. We must note that the *mecA* gene is shared between *Staphylococcus* species from both humans and animals, similarly to other antibiotic-resistance genes[58].

Crucially, natural transformation can transfer longer DNA fragments, such as SCC*mec* II (Fig. 4g), that are too large to be packed into the typical staphylococcal bacteriophages[19] and cannot be abolished by inactivating the donor, unlike other HGT mechanisms such as phage transduction, conjugation, and the staphylococcal pathogenicity island-helper phage system. In order to counter staphylococcal evolution by SCC systems, specific control methods against transformation are therefore necessary and biofilm formation can serve as a promising target.

## Methods

**Bacterial strains, plasmids, primers, and culture conditions**. Bacterial strains and plasmids used in this study are listed in Supplementary Table 5. The primers used in this study are listed in Supplementary Table 6. Clinical staphylococcal samples (99 MSSA isolates and 8 MR-CoNS isolates) were collected from the Kanto area of Japan. The reporter plasmids (pMK3-com-gfp, pRIT-com-gfp, or pHY-P$_{comG}$gfp-P$_{comE}$dsred), the sigH overexpression plasmids (pRIT-sigH or pRIT-sigHNH7), and the empty vectors (pHY300PLK or pRIT5H) were introduced into each strain by either electroporation or transduction[59] after passaging through strain RN4220.

For routine cultures, staphylococci were grown in TSB. For biofilm formation and transformation assays, S. aureus was grown in CS2 (complete synthetic medium, composition is shown in Supplementary Table 7)[19], TSB (tryptic soy broth), BHI (brain–heart infusion), RPMI 1640, M9 + aa (M9 medium supplemented with amino acids as in CS2), or MB (whole milk and BHI [1:1 ratio]). E. coli strains were grown in Luria broth (LB). Cultures were incubated at 37 °C either with shaking (180 rpm) or statically. Where required for selection, culture medium was supplemented with chloramphenicol (12.5 µg mL$^{-1}$), kanamycin (100 µg mL$^{-1}$), tetracycline (5 µg mL$^{-1}$), cefoxitin (4 µg mL$^{-1}$), cefmetazole (4 µg mL$^{-1}$), or ampicillin (for E. coli alone, 100 µg mL$^{-1}$).

**DNA extraction and genetic characterization of staphylococcal strains**. The staphylococcal cells were harvested from log-phase cultures and lysed by 0.1 mg mL$^{-1}$ lysostaphin before DNA was extracted using the standard phenol–chloroform method. Multiplex PCR was performed using the QIAGEN multiplex PCR kit to determine the CC types in 99 MSSA strains[30], to determine the SCCmec type in the MR-CoNS donors[60], and to test the presence of phages in S. aureus strains[61], using the corresponding primers in Supplementary Table 6. The presence of conjugative genes (nes, traA) was tested by PCR using the primer sets nesF/nesR and traAF/traAR, respectively[62] (Supplementary Fig.19a, b). The presence of plasmids in transformable strains was tested by electrophoresis of purified DNA as shown in Supplementary Fig. 19c. The absence of phage in donors (NefΔcls2-tet$^R$ and COLw/oφ) was also confirmed by phage susceptibility, using the plaque assay[59]. The presence of the blaZ and blaI genes was tested by PCR using the primer sets blaZF/blaZR and blaIF/blaIR, respectively. Long amplifications of SCCmec were performed using KOD One PCR master mix (TOYOBO).

The attB flanking regions were amplified by PCR, using primers orfXfor and unirev that encompass the attB region[63]. Direct sequencing of the PCR products was performed by Fasmac Co., Ltd. The sequence data were assembled and analyzed by SeqMan Pro 17 software (DNASTAR).

**Construction of dual-reporter plasmid**. The dual reporter of P$_{comG}$-gfp & P$_{comE}$-dsRed (pHY-P$_{comG}$gfp-P$_{comE}$dsRed, Supplementary Fig. 1) was constructed as follows. The clpB terminator (clpBter) and the promoter of the comE operon (P$_{comE}$)[19] were amplified by PCR from Nef using the primer sets clpBterF/clpBterR and PcomEF/PcomER, respectively. The dsRed was amplified by PCR from pJE-BAN6 plasmid[64] using the primers dsRedF and dsRedR. The three PCR products were ligated by overlapping PCR using the primers clpBterF and dsRedR before the resultant ligated product (clpBter-P$_{comE}$-dsRed) was digested by EcoRI and SalI and cloned into the corresponding site of pHY300PLK to generate the plasmid pHY-P$_{comE}$-dsRed. This plasmid was then linearized by inverse PCR using the primers pHYF and pHYR, eliminating the Amp$^R$ and its promoter region. The PCR product was digested at the primer-attached BglII and KpnI sites and ligated with the P$_{comG}$-gfp transcriptional fusion cassette, which was amplified by PCR from pMK3-com-gfp with primers PcomGF and gfpR, and digested by BglII and KpnI.

**Construction of deletion and substitution mutants**. Mutants were constructed by double-crossover homologous recombination using the pMADtet vector[19] (Supplementary Table 5). Briefly, two fragments flanking the upstream (primers A and B, Supplementary Table 6) and downstream (primers C and D, Supplementary Table 6) regions of the locus targeted for deletion (or substitution) were amplified by PCR using chromosomal DNA from Nef as template. The PCR products (AB and CD fragments) were used as template to generate the construct AD by overlapping PCR, using the primers A and D. Product AD was cloned into the BamHI–SalI site of pMADtet to generate the vectors (pMADtet-Δ3 to Δ17), pMADtet-ΔccrAB, pMADtet-ΔH, and pMADtet-attB* (Supplementary Table 5). In terms of attB substitution, primers B and C were designed not to change the coding amino acid sequence of OrfX. The plasmids, purified from E. coli, were introduced into Nef, N315 by electroporation, or into 9s by transduction after passaging through RN4220. Mutants (tetracycline sensitive, β-galactosidase negative) were selected[19] and the absence of the target gene was confirmed by PCR using the primers E and F (Supplementary Table 6). The attB substitution mutant was confirmed by restriction digestion (HindIII: included in the designed primers attB-B and attB-C) of the PCR product generated by primers E and F (Supplementary Table 6). NefΔcls2-tet$^R$ strain was created by transduction[59] using the donor N315Δcls2-tet$^R$ (carries a tetracycline-resistance gene at the cls2 locus[65]).

**Complementation of ΔTCSs**. For in-trans complementation, each TCS gene (including its own promoter) was amplified by PCR from Nef using the primer sets TCS12CF/TCS12CR, TCS13CPrF/TCS13CR, and TCS17CF/TCS17CR for TCS12, TCS13, and TCS17, respectively. The PCR products were cloned into the multiple cloning site of pHY300PLK (Takara) to generate the complementation plasmids pHY-12, pHY-13, and pHY-17. These plasmids were introduced into the corresponding mutants by electroporation after passaging through RN4220.

For chromosomal complemented of Δ13 and Δ17, each TCS and its flanking region were amplified by PCR using primers G and H (Supplementary Table 6). PCR product was cloned into BamHI–SalI site of pMADtet to generate the vectors pMADtet-TCS13 and pMADtet-TCS17. The plasmids were purified from E. coli and introduced into the corresponding ΔTCS by electroporation after passaging through RN4220. Complemented mutants (tetracycline sensitive, β-galactosidase negative) were selected[19] and the presence of the restored gene was confirmed by PCR using primers E and F (Supplementary Table 6).

**Growth curves**. Overnight cultures were diluted 1:200 in either TSB or CS2 and 200 µL was transferred to a 96-well flat-bottom microplate (Thermo Scientific), which was then incubated in a multimode plate reader (2300 Enspire™, PerkinElmer®) with shaking at 37 °C. OD$_{600}$ was measured every 30 min.

**Measurement of comG promoter activity by plate-reader assay**. To measure P$_{comG}$ activity by plate reader, reporter strains carrying the pMK3-com-gfp plasmid were grown overnight with 100 µg mL$^{-1}$ kanamycin and diluted 1:200 in either CS2 or TSB supplemented with vancomycin, bacitracin, or nisin as appropriate. Next, 200 µL of these diluted cultures were placed in a transparent, 96-well flat-bottom microplate (Thermo Scientific) before continuous incubation (with shaking) at 37 °C in the multimode plate reader (2300 Enspire™, PerkinElmer®). Changes in the RFU (relative fluorescence units) and OD$_{600}$ were measured at 30-min intervals. The RFU was normalized by the OD$_{600}$ value.

**Fluorescence microscopy**. For planktonic conditions, to observe and determine the percentage of GFP or dsRed-expressing cells in planktonic conditions, 50 µL of overnight culture for each reporter strain was inoculated into 10 mL of CS2 medium in a glass vial. These cells were grown at 37 ˚C with shaking for the appropriate time period before 5 µL of the culture was placed on slide, sealed with a cover glass, and observed by the fluorescence microscope (BZ-X710, Keyence). The total number of counted cells is indicated in figure legends. The percentage of GFP or dsRed-expressing cells was calculated by dividing the number of fluorescence-expressing cells by the total number of cells.

For biofilm conditions, overnight cultures of P$_{comG}$-gfp reporter strains were diluted 1:200 in TSB and grown with shaking for 3 h at 37 °C. Cells were harvested from 750 µL of the culture, washed and suspended in 1.5 ml of CS2, and transferred to a well of a flat-bottom 6-well polystyrene plate (Costar®, Corning). These plates were incubated statically for up to 4 days at 37 °C. Bacteria in the biofilm were collected by extensive pipetting, washed and suspended in 1xPBS, and stained by propidium iodide (40 µM final concentration) (WAKO) to distinguish dead cells (red fluorescence) from living cells. Stained bacteria were observed by the fluorescence microscope. At least 300 living cells were counted in every independent experiment. The percentage of GFP-expressing cells was calculated by dividing the number of GFP-expressing cells by the total number of living cells. None of the detected values in the fluorescence microscopy were assigned the value 0 for the calculation of mean values and statistical analyses.

**Antimicrobial-susceptibility testing**. Minimum inhibitory concentration (MIC) assays (Supplementary Table 1 and 4) were conducted in a 96-well microtiter plate (round bottom). Overnight bacterial cultures were diluted 1:2000 in appropriate medium and 100-µL aliquots (~5 × 10$^8$ CFU) were used to inoculate wells containing cation-adjusted Muller– Hinton broth (MH), TSB, or CS2 media supplemented with twofold serial dilutions of antibiotics (vancomycin, bacitracin, nisin, cefmetazole, or cefoxitin). The plates were statically incubated for 20 h at 37 °C. The MIC was determined by the lowest concentration of antibiotic at which growth was inhibited.

Disk-diffusion testing was conducted according to CLSI standards[66] using the direct colony-suspension method. Glycerol stocks of 9s, 35s, CoNS15, and 35s[CoNS15] were streaked on drug-free TSA. Unstable transformants (9s[CoNS15]) were streaked on TSA supplemented with 4 µg mL$^{-1}$ cefoxitin. Emerged colonies were suspended in 0.85% NaCl and turbidity was adjusted to 0.5 McFarland standard. The inocula were swabbed on Mueller–Hinton agar and the antibiotic disks of oxacillin (1 µg), and cefoxitin (30 µg) (KB disks, Eiken Chemical) were used for susceptibility testing. Zones of inhibition were determined following 18 h of incubation at 35 °C.

**Biofilm staining and quantification**. Biofilm formation was assessed in flat-bottom 96-well polystyrene plates[67] (Supplementary Figs. 5 and 10–12), or in 6-well plates (Supplementary Fig. 5b).

For 96-well plates, overnight cultures were diluted 1:200 in the medium to be tested, and 200 µL were transferred to each well. Following 24 h of static incubation at 37 °C, the nonadherent cells in medium were aspirated and the wells were stained with 200 µL of 0.1% crystal violet for 15 min. The wells were then gently washed 3 times with 200 µL of 1xPBS to remove residual stain before air-drying.

For biomass quantification, 100 µL of 96% ethanol was added to the wells and incubated at room temperature for 10 min to solubilize the stain. The absorbance at 595 nm ($A_{595}$) of the resolved stain was measured by the plate reader. Cell-free wells were used as blanks.

For 6-well plates, overnight cultures were diluted 1:200 in TSB and grown with shaking for 3 h at 37 °C. Cells were harvested from 750 µL of the culture, washed and suspended in 1.5 ml of CS2, and transferred to a well of the flat-bottom 6-well polystyrene plate. These plates were statically incubated for up to 4 days at 37 °C. Every 24 h, formed biofilms were stained and quantified. Biofilm staining and quantification was carried out as described above using larger volumes of solutions (1.5 mL of 0.1% crystal violet and 750 µL of 96% ethanol). Cell-free wells were used as blanks.

**Preparation of donors for natural transformation assays.** Donors used in natural transformation assays include either a purified plasmid (pHY300PLK)[59] (Supplementary Fig. 8 alone) or heat-killed cells. For heat-killed tetracycline-resistant donors, N315Δcls2-tet$^R$, NefΔcls2-tet$^R$, or Nef-pT181 were used. The former two carry a tetracycline-resistance gene in the chromosome (cls2 locus), while Nef-pT181 carries a tetracycline-resistance gene in plasmid pT181.

Heat-killed donors were prepared by diluting an overnight culture 1:20 in TSB and growing with shaking for 3 h at 37 °C. Next, cells were harvested and suspended in 5 ml of 1xPBS before heating in boiling water for 10 min. The absence of viable cells was confirmed by plating on TSB agar.

**Natural transformation in planktonic conditions using purified plasmid donor.** Natural transformation assay to test the effect of nisin (Supplementary Fig. 8) was carried out in planktonic conditions using purified plasmid DNA[19] as follows. About 500 µL of recipient cells from overnight cultures were washed and inoculated in 10 ml of CS2 and grown for 2 h at 37 °C with shaking. Then, 8 µg mL$^{-1}$ nisin was added to the medium as appropriate and further grown for 8 h. Cells were then harvested and suspended in fresh 10 ml of CS2 containing 10 µg of purified pHY300PLK plasmid and growth continued for an additional 2.5 h before pouring into melted BHI agar supplemented with 5 µg mL$^{-1}$ tetracycline to select for transformants. In this case, we chose BHI as a complex, rich medium although other media may be viable for this purpose. The plates were incubated for 2 days until colonies emerged. NefΔcomE strain was used as a negative control.

**Natural transformation in planktonic conditions using heat-killed donor cells.** For natural transformation assays in planktonic condition using heat-killed cells as a donor (Fig. 2c), recipient cells from 500 µL of overnight cultures were washed by CS2 and inoculated in 10 ml of CS2 with $5 \times 10^{10}$ CFU-equivalent heat-killed N315Δcls2 cells. The ratio of heat-killed donor ($5 \times 10^{10}$ CFU equivalent) to recipient (before transformation assay, ~$2 \times 10^{10}$ CFU) was approximately 2.5. Cells were grown at 37 °C with shaking for 3 days before pouring into melted BHI agar supplemented with 5 µg mL$^{-1}$ tetracycline to select for transformants. The plates were incubated for 2 days. ΔH strain was used as a negative control.

**Natural transformation in biofilms.** To detect natural transformation in biofilms (Figs. 2 and 3, Supplementary Fig. 9–13), overnight cultures of recipient cells were diluted 1:200 in TSB and grown with shaking for 3 h at 37 °C. Cells were harvested from b 750 µL of the culture, washed, and resuspended in 750 µL of appropriate growth medium (CS2, TSB, BHI, MB, RPMI 1640, or M9 + aa). This 750 µL of cell suspension (~$10^8$ CFU mL$^{-1}$) was then distributed in 6-well polystyrene plates and 250 µL of heat-killed donor cells (~$10^9$ CFU mL$^{-1}$) was added. The total volume of growth medium was adjusted to 1.5 mL per well. The ratio of heat-killed donor (~$2.5 \times 10^8$ CFU) to recipient (before transformation, ~$7.5 \times 10^7$ CFU) was approximately 3. The 6-well plate was statically incubated at 37 °C and the medium was refreshed every 24 h. After incubation for an appropriate time period, the biofilm was harvested by extensive pipetting and poured into melted BHI agar supplemented with 5 µg mL$^{-1}$ tetracycline or 4 µg mL$^{-1}$ cefmetazole, depending on the donor used. The plates were incubated for 2 days.

NefΔcomE or ΔH strains were used as negative controls as indicated in the figures. Heat-killed Nef was used as a negative-control donor since it lacks the tetracycline and the mecA genes.

**Confirmation of transformants in the biofilm-transformation assays.** For tetracycline selection, to confirm the acquisition of resistance, generated colonies were replicated on 5 µg mL$^{-1}$ tetracycline plate. Tetracycline at this concentration does not allow emergence of spontaneous resistant colonies[27].

To conduct the PCR tests, whole DNA from the replicated colonies was extracted following growth in liquid cultures (Supplementary Fig. 9). The presence of the tetracycline-resistance gene was confirmed by PCR using the primers TetF(salI) and TetR(EcoRI).

The presence of pRIT5H plasmid in the transformants derived from Nef-pRIT5H recipient was confirmed by PCR using the primers pRIT5H1 and pRIT5H2 (Supplementary Fig. 9). Spontaneous tetracycline- resistant mutants did not emerge in any recipient strain in these assays as confirmed by using a heat-killed Nef donor (Supplementary Fig. 9, 13c).

For cefmetazole selection, to confirm the acquisition of resistance, generated colonies were replicated on fresh 4 µg mL$^{-1}$ cefmetazole plates (Supplementary Fig. 14). Colonies that could grow on the replica plates underwent single-colony isolation, growth in liquid culture (to avoid any contamination of donor DNA), and genomic extraction (Supplementary Fig. 14). The presence of the mecA gene was tested by PCR using the primers mecAF and mecAR.

To confirm that the mecA PCR signal is not derived from the donor DNA remaining outside of the cells, DNase I and PMA were utilized (Supplementary Fig. 15). PMA is a photoreactive dye that binds covalently to DNA and inhibits amplification by PCR. DNase I and PMA cannot penetrate into living (membrane-intact) cells.

For DNase-I treatment and PCR, overnight cultures were diluted 1:200 and were grown in TSB supplemented with or without cefmetazole. At OD$_{600}$~1, 400 µL cultures (~$10^8$ CFU) were harvested, washed, and resuspended in fresh TSB. To prepare heat-killed N315 controls, the resuspended cells were heated in boiling water for 10 min. Cells in TSB were treated with (or without) 50 µg mL$^{-1}$ DNase I at 37 °C for 3 h, followed by DNA extraction. PCR for mecA and gyrA was carried out using the primer sets mecAF/mecAR and gyrAF/gyrAR, respectively.

For PMA–PCR, two sets of 400 µL of culture (~$10^8$ CFU) were prepared as described in the section of DNase-I treatment. One set was heated in boiling water for 10 min, while the other was left untreated. Both sets of culture were then treated with (or without) 50 µM PMA (Biotium, CA) according to the manufacturer's instructions. Briefly, the samples were incubated with (or without) PMA in the dark for 10 min, followed by exposure to blue-light-emitting diodes (LEDs) for 15 min. The cells were then pelleted by centrifugation and the DNA was extracted before PCR for mecA and gyrA was carried out.

To confirm that our transformants of S. aureus (mannitol positive) were not due to contamination from a CoNS donor (mannitol negative), they were streaked on mannitol salt agar and incubated at 30 °C for 18h (Supplementary Fig. 18a).

**Calculation of transformation frequency.** Transformation frequency was calculated as the ratio of the number of transformants to the total CFU after transformation. Nondetected values were assigned half the value of the detection limit of the strain for the calculation of mean values and statistical analyses.

**Stability test for cefoxitin resistance.** Glycerol stocks of mecA transformants were streaked on BHI plates supplemented with 4 µg mL$^{-1}$ cefoxitin. Emerged single colonies were inoculated into BHI medium supplemented with 4 µg mL$^{-1}$ cefoxitin and grown with shaking at 37 °C for 12 h. This culture was then diluted 1:1000 in drug-free BHI medium and grown for another 12 h. After 12 h, cells were plated on drug-free BHI agar and were replicated on BHI agar supplemented with 4 µg mL$^{-1}$ cefoxitin to assess the population percentage that maintained growth ability under selective pressure from the β-lactam drug.

**Genome sequencing and sequence analysis.** Genome sequencing of 9s, 35s, CoNS15, 9s[N315], 35s[CoNS15], and 35s[N315] was carried out via an Illumina Novoseq 6000 platform (Novogen, China). Reads were assembled de novo using SeqMan NGen 17 (DNASTAR). The resultant assembled contigs were annotated for each gene using SeqBuilder Pro 17 (DNASTAR). Transformant contigs containing SCCmec were aligned against recipient and donor contigs or the N315 genome (BA000018) by Megalign Pro 17 using MAUVE (DNASTAR).

**Statistics and reproducibility.** Statistical analyses were performed by GraphPad Prism (GraphPad Software, version 8.4.3) on data from three or more independent experiments. Error bars indicate SD for three or more independent experiments. The differences between groups were analyzed by one-way ANOVA followed by Dunnett's or Tukey's multiple-comparison test as indicated in figure legends. The log values of natural transformation frequencies were analyzed statistically. $^*P < 0.05$, $^{**}P < 0.01$, $^{***}P < 0.001$, $^{****}P < 0.0001$ were considered statistically significant. The full statistical results are provided in the Source data file. All experiments in Fig. 1a, b; 2a-c, f and Supplementary Figs. 1b, c; 2; 3; 4b, c; 7; 10a; 11a; 12a, b; 13b, c. were repeated at least three times independently. All experiments in Fig. 4a–c, e–g and Supplementary Figs. 5a, c; 6a, b, c; 8; 9a, b; 10b; 11b; 12c; 13a; 15a, b; 16a, b; 18a, b. were repeated at least twice independently with similar results.

**Reporting summary.** Further information on research design is available in the Nature Research Reporting Summary linked to this article.

## Data availability

The genome-sequence data generated in this study have been deposited in the NCBI Sequence Read Archive under accession code SRP340166 The S. aureus N315 genome sequence used in this study is available in the NCBI GenBank database under accession code BA000018 Other relevant data are provided within the paper and its Supplementary Information. Source data are provided with this paper.

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

## Acknowledgements
This study was supported by Takeda Science Foundation, The Waksman Foundation of Japan, Pfizer Academic Contributions, JSPS KAKENHI Grant Numbers 25860313 and 18H02652, JSPS Bilateral Open Partnership Joint Research Projects JPJSBP120199916 (to KM), and Program to Disseminate Tenure Tracking System, MEXT (to RLO). We would like to express our appreciation to Dr. Teruyo Ito for valuable discussion, and Ms. Yoshimi Tsutsumi, Mr. Tin Ming Tan, Mr. Bobby Sookhoo, Ms. Shenghe Huang, and Ms. Clara Effenberger for their experimental help. We would also like to thank Dr. Bryan J. Mathis, Medical English Communications Center, University of Tsukuba, for language revision of this paper.

## Author contributions
T.M. and K.M. conceived the study and supervised the research. M.M., L.T.T.N, and R.L.O. contributed to the study design. M.M., L.T.T.N., R.L.O, and M.H. performed experiments, and analyzed data. M.M. and K.M. wrote the paper with all other authors.

## Competing interests
The authors declare no competing interests.

## Additional information

**Peer-review information** *Nature Communications* thanks Maria Miragaia and the other, anonymous, reviewers for their contribution to the peer review of this work. Peer reviewer reports are available.

