## [Peer Review File · Nature Communications]

Reviewers' Comments:

Reviewer #1:

Remarks to the Author:

Nature Communications Revision

General Comment

In previous studies the same authors showed that *S. aureus* natural transformation was sigH-dependent and occurred only in the presence of an artificially synthesized media (CS2). These first studies were performed in planktonic conditions and the efficiency of transformation was extremely low.

In this study authors aimed to prove that in *S. aureus* natural transformation is more common in biofilms than in planktonic cells. Moreover, they intended to show that SCCmec was frequently transferred in biofilms by natural transformation. The findings that SCCmec is transferred in biofilms are new although it is not new that transfer of mobile genetic element occurs in biofilms. The knowledge produced is of high interest for the scientific community working with *Staphylococcus* and could be of a wider interest because it could serve as a model of how bacteria not considered being transformable can in certain specific conditions become transformable. Statistical analysis is adequately performed and methodology is generally described with sufficient detail.

To prove their hypothesis, authors constructed mutants derived from the N315 MRSA strain lacking phage PhiN315 and SCCmec (Nef) and that expressed sigH (Nefh), as well as several MSSA clinical strains to serve as recipients; and purified DNA and heat inactivated cells of different MRSA and MR-coagulase-negative strains to serve as donors. Also they constructed several two-component system (TCS) and comG and comE deletion mutants to test the impact of these genes/systems in the capacity of the recipient cells to incorporate exogenous DNA. Both planktonic and biofilm states were tested.

The focus of the study – the mechanism of SCCmec transfer – is still an open and key question in methicillin-resistant *S. aureus* (MRSA) biology, with implications in infection control and treatment of infections caused by these bacteria. I congratulate the authors for the gigantic effort and the immense amount of work done to prove their hypothesis. However, the study presented is not totally convincing mainly due to methodological issues that require further clarification. Also some of the conclusions are overstating and do not completely agree with the results shown.

Major comments

- i) The assays of transformation in planktonic cells and in biofilms, were done with different proportions of donor DNA vs recipient cells and for that reason efficiencies of transformation in these two situations are not directly comparable;
- ii) Transformation assays were still done in very artificial conditions, using mostly synthetic media, adding cell wall targeting antibiotics, and using a high amount of donor DNA, which is far from the real situation. It is thus difficult to convince that the phenomenon of transformation shown in the paper is in fact natural.
- iii) Is not clear if primers used for checking the integration of SCCmec in the recipient strains upon transformation target only the SCCmec or also the integration region in the recipient. This is crucial to understand if the SCCmec was actually integrated in the chromosome of the recipient.
- iv) It is not clear how the authors dealt with the possible contamination of the recipient cells with donor DNA after mixing. For example addition of DNase after mixing donor heated inactivated cells with recipient could have avoided possible contaminations. In a PCR reaction any contamination even if in low amount would come out as a positive result. How did authors guaranteed that PCR amplification of *mecA* and SCCmec element regions correspond to DNA inside the recipient cells and not to contamination with the donor DNA? There are several proteins at cell surface that have DNA binding capacity, like Atl, that could have recruited donor DNA to the surface of the recipient cell.
- v) The MSSA and MRSA strains used as recipients and donors were not characterized for their

genetic background. The few isolates for which the genetic background was characterized all belonged to CC5, which is the same genetic background as N315 strain. The CC5 genetic background was previously described to have a higher recombination rate and higher genetic diversity than the remaining *S. aureus* clonal complexes, and transformation ability might be a characteristic of this specific clonal lineage. It is thus difficult to evaluate how global is the phenomenon of transformation in the entire *S. aureus* population.

vi) Although transformation in *S. aureus* was achieved, apart from the expression of *comG* no clear molecular evidences were provided that a competent state was induced, that other competence genes in the genome were involved and that donor DNA interacted with the *com* genes during this process. Complementation of the results obtained with expression data, DNA-protein binding assays, and/or co-localization experiments of DNA and *com* genes using microscopy or other approaches would help to better sustain your results.

vii) The original N315 isolate, besides containing the phage Phi-N315 also contains at least one plasmid (pN315), however authors do not mention whether the strain used as donor was cured from this plasmid. This is important because plasmids can also be involved in SCCmec transfer. This observation is also valid for the other strains used as donors, including MRSA and MR-CoNS. Did the authors determine the whole genome of these strains to make sure they did not contain phages or conjugative plasmids? This is important because transduction and conjugation could have been used instead of transformation as mechanism of SCCmec transfer.

viii) Another characteristic that was shown to be important for the success of the SCCmec transfer is the presence of *blaZ* and its regulators. Do the authors know if the recipient strains tested contained these genes?

ix) Although I agree that authors provided evidences that SCCmec can be transferred by transformation, the efficiency of this process appears to be very low. When clinical strains were used as recipients only one (35s) was able to stably maintain SCCmec. I do not think that authors provided enough evidence to support that transformation is the main mechanism of SCCmec transfer.

x) The hypothesis raised by this study is that SCCmec can be transferred by natural transformation in biofilms. This implies that emergence of MRSA occurs during infection in polymicrobial biofilms. In what particular clinical conditions do you preview this might occur? This can be included in the Discussion section.

xi) Limitations of the study should be included in the Discussion section.

Specific comments

Page 1, Title,

The title is misleading; MRSA has been shown to emerge by other transfer mechanisms besides transformation, like transduction and conjugation. Also although authors showed that SCCmec was transferred by transformation, this was achieved in very artificial conditions, so I wonder if "natural transformation" is the more adequate term. Please change the title to accommodate these ideas.

Page 2, Introduction, line 43-45:

Please verify if MRSA is the leading cause of nosocomial infection. Antibiotic-resistant Gram-negative bacteria nosocomial infections are increasing and are now a major concern. In the community CA-MRSA is also decreasing and LA-MRSA is not very frequent in human infection.

Page 2, Introduction, line 46:

Specify the type of percentage you are talking about. Is it of nosocomial infection? Or other?

Page 2, Introduction, line 53-55

Methicillin resistant determinant was found outside SCCmec in some coagulase-negative staphylococci, like *S. fleurettii* and some *S. vitulinus*. In these species *mecA* it is located in the chromosome, outside any mobile genetic element. Please rephrase.

Page 2, Introduction, line 56

Indicate what do CcR stands for. Also the function of *orfX* was already described and the name of the gene was changed. Please update this information.

Page 3, Introduction, line 68-70

Add that transformation was achieved only in the presence of a particular growth medium (CS2).

Page 3, Introduction, line 77

Add information on other studies wherein SCCmec transfer was accomplished using other mechanisms such as transduction and conjugation.

Page 3, Results, line 78

It is not obvious why did the authors looked in the first place into TCS in the context of natural transformation in *S. aureus*. This has to be contextualized. Has this association been done before in other bacteria? Why were deletion mutants constructed for this type of system in particular? Some information is missing here.

Page 4, line 95

How do authors know that the Nef strain does not contain any conjugative element or lysogenic phage? How did authors dealt with genes with unknown function in this strain?

Page 4, line 99

Can you please provide the composition of the CS2 medium? This is important to understand if any component in particular can be inducing the competent state. What this medium has that is so crucial for the occurrence of transformation? In the Lab, when preparing competent cells it is important to add CaCl₂ so that DNA binds to cell surface. Is there any component doing the same function in CS2 medium? If this is the case I do not think that authors can say this is a natural phenomenon.

Page 4, line 104

I do not think that growth defects is the correct term to use, since authors did not look into cell morphology. What authors are probably observing is cell lysis in the stationary phase. Bacteria grow and divide mainly in the exponential phase what is not frequent in the stationary phase.

Page 4, line 105

Please indicate to what specific yield are you referring to. Do you mean higher OD?

Page 5, line 113-114

Authors refer in the text that complementation in trans within these delta-TCSs mutants restored the percentages of GFP-positive cells however they do not provide the figures showing this. Please include microscopy images corresponding to the % of GFP-positive cells for the complemented strains.

Page 5, line 116

With the results showed up until this point the authors can only affirm that TCS12, TCS13, and TCS17 are involved in the regulation of the *comG* not the entire competence machinery. Competence is a cell state provided by a plethora of genes not the expression of a single gene. Actually later in the paper authors reach to the conclusion that *comG* is not essential for natural transformation, which is somehow contradictory.

Page 5, line 121-122

What were the breakpoints used by the authors to consider a strain as being resistant or susceptible to vancomycin, bacitracin and nisin? Clinical breakpoints are not defined for TSB or CS2 medium.

Page 5, line 134-137

When treating cells with vancomycin and bacitracin the bacterial growth may be affected as well as global gene expression. Could it be that the decrease in expression is not specific of comG, but could be a global decrease in gene expression due to a decrease in growth? This could be overcome if the expression of a control gene not involved in competence would have been used or by showing that the growth rate did not change in the presence of antibiotics.

Page 6, line 136-137

I am not sure that authors can affirm that there was an increase in comG-GFP reporter expression when cells were exposed to nisin. Is the increase observed at 8h statistically significantly different from that observed at 4 ug/mL? If the difference is not significant then nisin has an effect similar to that of vancomycin and bacitracin, decreasing the comG-GFP expression with the increase in the concentration.

Page 6, line 143

Authors say that the percentage of GFP-expressing cells in biofilm was less than in planktonic culture, and that data indicates that the Agr quorum sensing TCS13 plays a role in comG reporter expression. However, they do not specify what data are they taking about. Please specify.

Page 6, line 156

Please clarify if tetracycline resistance gene in the donor strain was encoded in the chromosome or in a plasmid and what was the size of this plasmid.

Page 6, line 158

Authors showed that in Nefh, the transformation frequency was similar to Nef under biofilm-forming conditions. These results and results obtained with clinical strains suggest that in biofilms transformation might be independent on sigH. This would represent a totally new mechanism different from the one described previously by the authors. This should be clear in the paper.

Page 7 line 167-169

Unexpectedly, authors found that the negative control (deletion mutant for comG) was transformable, however, all the experiments performed previously used this gene as a reporter for the occurrence of transformation. How do the authors conceal the previous results with this observation?

Page 8, line 189-190

The influence of genetic background in the ability of strains to stably maintain the SCCmec was previously reported. The knowledge on the genetic background of transformable and non-transformable strains should be provided, because this would indicate if the phenomenon of transformation described here is specific of certain genetic backgrounds and conditions or a more global occurrence in *S. aureus*.

Page 8, line 197-198

It is possible that SCC elements are transferred through non-conjugative plasmids if conjugative plasmids are also present in the same strains or if the plasmid is transferred within a phage. A SCC element containing mecB was already found in *S. aureus* in a plasmid, this has also been described in *Micrococcus caseolyticus*, a genera very closely related to *Staphylococcus*, which appear to be involved in the emergence and evolution of SCCmec. Please rephrase to include this information.

Page 8, line 204

More information should be provided on the 20 MSSA strains used as recipients. To what genetic background do they belong? This is important information because not all the isolates were transformable and the genetic background was reported to have a role on genetic transfer of SCCmec.

Page 10, line 235-236

The impact of attB (chromosome) and attS (SCCmec) sequences and adjacent sequences in the integration of SCCmec by Ccr has been previously reported (Wang et al. 2012) and could eventually explain the lack of stability of SCCmec in the recipient strains. Have the authors checked the diversity in the attB and attS sequence of recipient strains and SCCmec? Differences in the efficiency of integration in the chromosome can also derive from difference in methylation patterns and R/M systems or the presence of CRISPR in the recipient strains? The results obtained regarding stability should be discussed in these contexts.

Page 10, line 236

Authors should clarify what kind of primers did they use (targeting what) to detect the presence of SCCmec in the recipient strain upon transformation. The PCR should have been designed to amplify the junction site between recipients DNA and the SCCmec and sequencing done to confirm. If the PCR reaction only targeted the mecA or fragments of the SCCmec, amplification could correspond to contamination of the recipient DNA with donor DNA.

Page 11, line 263

Please clarify which was the genetic determinant that was transformed into the recipient strain when transformation was tested in biofilms in the presence of the cell wall-targeting antibiotics.

Page 12, line 287

I think authors meant SCCmec I-IV instead of SCCmec I-V?

Page 13, line 311

Proposing that natural transformation in biofilms is the main means of SCCmec transfer is overstating. The conditions described in this study are somehow artificial (CS2 medium) and assume that in the biofilm total DNA of the donor is available. Moreover, there is no direct comparison with other transfer mechanisms.

Page 13, line 312

The transformation efficiency occurring in the natural environment should be largely decreased when compared to the laboratory assays of this study, because the growth media will not be CS2, the concentration of DNA of donor would be lower and compatibility of donor and recipient DNA must exist so that SCCmec is acquired and maintained.

Page 13, line 319-321

Authors cannot affirm that they tested the impact of genetic background on the efficiency of transformation because they did not characterize the genetic backgrounds of the donors and recipient strains.

Page 13, line 323-324

In my opinion the assays performed do not allow for a direct comparison of the efficiencies of transformation between the planktonic and biofilm conditions because the conditions in planktonic and biofilm were not exactly the same. In particular the number of recipient cells and amount of donor DNA were not normalized in the two assays.

Page 14, line 333-335

This sentence is not clear. Please rephrase.

Page 14, line 339-342

Please elaborate more on the relation between nuclease heat inactivated cells and transformation occurring in biofilms. This is not very clear.

Page 14, line 343-344

Please explain better the relation between eDNA sequestration in biofilms and the fact that genomic purified DNA cannot serve as a transformation donor but heated inactivated cells can. Do authors think that donor proteins or metabolites might be important to induce the competent state

in the recipients?

Page 15, lines 357-358

It is not proven that in colonization state *S. aureus* form biofilms. Usually the cfu load is low during skin or mucous membranes colonization. In fact Krismer et al (2011) point towards a dispersed rather than a biofilm-related mode of growth during *S. aureus* nasal colonization.

Page 19, lines 461

Do $5 \log_{10}$ stand for cells or for CFU/mL?

Page 32, Figure 3 Legend

Figures are of very poor quality. Please try to produce ones with better resolution in which the bands are more visible, including those of the marker.

Supplementary Table 4

Add information about the units (CFU/mL?) and what it means the number in parenthesis.

General comment: In the description of the transformation experiments, including the Figure legends, it is almost always missing the strain that was used as donor. Also it is missing the information if heat-killed cells or purified genomic DNA was used. This is important because the donor and type of DNA can also have an impact in the outcome of transformation. Methylation pattern and R/M systems might have a role. Please go through the text and add this information where is missing.

Reviewer #2:

Remarks to the Author:

The manuscript by Maree et al., is dedicated to investigate natural transformation in *Staphylococcus aureus*. The study is divided in two parts. In the first part the authors investigated how the presence of antibiotics, two-component systems and biofilm lifestyle affect the activity of sigH transcription factor. In the second part, the authors explored the transferability of the SCCmec element from heat-killed bacteria to different recipients in biofilm conditions. The biological question that the authors are trying to address is important and the manuscript lists several interesting observations related to the capacity of *S. aureus* to gain foreign DNA.

Major comments

- Three two-component systems (AgrCA, BraSR, and VraSR) out of the sixteen present in the genome of *S. aureus* were identified as important for the activity of SigH. AgrCA and BraS induce whereas VraSR represses the activity of sigH. Because the regulon of each of these TCS includes dozens of genes, it would be important to know which is the contribution of the modification of SigH expression to the DNA uptake phenotype of AgrCA and BraSR mutants. Can constitutive overexpression of SigH complement the DNA uptake phenotype of BraSR and AgrCA mutants?
- The results showed that DNA uptake occurs more efficient in biofilms than in planktonic cells. At the same time the authors found that mutants in AgrCA and BraSR displayed a reduced capacity to form biofilm. However, the authors have not explored how biofilm formation and DNA uptake phenotypes are related with BraSR, AgrCA activity. One possibility to explore this likely connection will be to identify a medium (TSB, BHI, M9) where BraSR and AgrCA mutants retained the capacity to produce biofilm. Then, the authors could explore the efficiency of DNA uptake of BraSR and AgrCA mutants in biofilms.
- The presence of subinhibitory concentration of vancomycin or bacitracin affect the activity of SigH. The effect of these antibiotics on cell wall is pleiotropic and they might not be any connection between these observations and BraSR and AgrCA mutants deficiency in DNA uptake and biofilm formation.
- The results showing that the SCCmec element can be transferred from heat-killed bacteria to different recipients are very interesting. I assume that BraSR and AgrCA mutants were not used to investigate the capacity to accept the SCCmec element because they are unable to form biofilm in CS2 medium. Again, a medium where BraSR and AgrCA mutants retain the capacity to form biofilm could allow to test their role in SCCmec transfer. Alternatively, the capacity to produce biofilm in vitro is very different among *S. aureus* strains. Thus, it is very likely that mutation of

BraSR and AgrCA does not affect the capacity to produce biofilm in CS2 medium in other *S. aureus* strains.

- In relation with the previous question, *S. aureus* is able to produce a biofilm matrix made of proteins or exopolysaccharides. Does the nature of the biofilm matrix affect the transformation efficiency?

Minor comments

The strain NefH is not described until line 158, whereas it is used in all the panels of Figure 2 that it includes experiments described before. Thus, either the strain is described earlier in the text or it is described in the legend of Fig. 2.

Reviewer #3:

Remarks to the Author:

The manuscript describes conditions believe to be natural transformation and builds of previous work showing that *S. aureus* can be transformable under special conditions where the com locus is expressed.

There are several major concerns with this manuscript. 1) In description of the central transformation experiments it is not indicated which DNA is being used in the experiment; at which antibiotic concentration were the transformants selected and on which agar plates. In some cases, it is also unclear which recipient strain was being used. In Materials and Methods it appears that at least in some cases the transformants are incorporated in BHI agar - why is performed and why BHI? 2) The transformants are not checked thoroughly. It is not confirmed whether the transformants are spontaneous resistant mutants arising or whether the actual DNA being used for the transformation is present in the recipient strain background. The transformants have to be genome sequenced to demonstrate that they actually is carrying the transformed DNA and that they have not picked up any phages in the process - see below 3) It is not clear what the Nef-delta-cla2-tetR strain is being used for but it has been made by transduction and we know that in transduction the majority of the transductants are carrying the transducing phage (commonly phi11 or 80a) in addition to phages from the donor strain 4) the transformation efficacy is reported as frequencies (for example supplementary table 4) but the actual number of colonies on the agar plates should be reported, 5) In figure 2 transformation with control DNA is missing not carrying the resistance genes that is selected for? 6) In the critical transformation experiments of the SCCMec cassette there is also a need for genome sequence analysis rather than PCR analysis of transformants in order demonstrate the transfer.

In general, the authors are doing very little to control for a number of artefacts that are obvious but will have to be accounted for particularly with the controversial results communicated. Further the authors are using a lot of space to monitor expression of comG under different conditions (all of figure 1 and figure 2a and b) but subsequently discard comG as being required for transformation (figure 2g).

Line 57-59: This is wrongly stated as the cited publication inferred the number of transfer events based on modelling

line 92: what do you mean by delta3-delta17? if you refer to the mutants why is nr. 2 missing?

line 99: What characterizes the CS2 medium?

figure 1c: How many cells were counted in each experiment?

line 141: Not clear where this is going and only comment in day 2 and 3 and when decrease on day 4

line 118-152: Is mostly based on supplementary information and seems very extensive to demonstrate very few points - can be shortened substantially

line 155: The logic is not clear as you in line 142-143 state that you have less comG expression in biofilms

line 156: what is the delta *cls2* and why is that introduced here?

line 158: why now introducing *sigH* overexpression - may be remembered from the introduction but could aid the reader if shortly mentioned why and also how this is overexpressed.

line 155-158: This had to be explained much better - what is the assay exactly and what are you transforming with and what are you selecting for?

line 158: Compared to what? in supplementary figure 6 it appears that there is an effect of *sigH* but not of *nisin*.

Figure 2c: Which type of DNA are you transforming with here, what are you selecting on and which strain is the recipient?

line 166: This is very suspicious as your initial assumption was that *comG* is needed for competence and also the logic of the manuscript becomes obscure as all the optimizations of conditions (figure 1) were done with *comG* as reporter

line 155-178: Where are the controls? There is a need for transformation with DNA lacking resistance genes and also genomic verification that the transformants have the recipient backbone but have acquired the transformed DNA.

line 184: What is the *sigH* expression level of the strain *r59* compared to strains that are not transformable without artificial *sigH* overexpression?

line 189: Also in these experiments are the controls missing.

line 204: You will need to show how the plates look with your transformants and also provide information on the number of transformants to demonstrate how well did cefmetazole work as selective agent.

line 222-225: how do you control for the recipient being the correct strain? Transformants should be genome sequenced to demonstrate that they carry the backbone of the recipient in addition to the entire SCCMec cassette.

line 228 and figure 3a: It is strange that you get weaker PCR bands for some of the transformants as you are not doing a quantitative PCR so the amount of template DNA should not be reflected in the PCR reaction.

line 229: Very surprising you see equal transformability with coagulase negative staphylococci where there there should be restriction-modification barriers

line 236-241: Again you should provide genome sequence information rather than these fluffy PCR bands.

line 253: *attB* is not a commonly used term for SCCMec integration, is it the putative integration site in *orfX*? you need to reference where the *attB* sequence was described and provide the sequence.

figure 4: Again, what was transformed with what?

Point-to-point response

Reviewer #1 (Remarks to the Author):

General Comment

In previous studies the same authors showed that *S. aureus* natural transformation was sigH-dependent and occurred only in the presence of an artificially synthesized media (CS2). These first studies were performed in planktonic conditions and the efficiency of transformation was extremely low.

In this study authors aimed to prove that in *S. aureus* natural transformation is more common in biofilms than in planktonic cells. Moreover, they intended to show that SCCmec was frequently transferred in biofilms by natural transformation. The findings that SCCmec is transferred in biofilms are new although it is not new that transfer of mobile genetic element occurs in biofilms. The knowledge produced is of high interest for the scientific community working with *Staphylococcus* and could be of a wider interest because it could serve as a model of how bacteria not considered being transformable can in certain specific conditions become transformable. Statistical analysis is adequately performed and methodology is generally described with sufficient detail.

To prove their hypothesis, authors constructed mutants derived from the N315 MRSA strain lacking phage PhiN315 and SCCmec (Nef) and that expressed sigH (Nefh), as well as several MSSA clinical strains to serve as recipients; and purified DNA and heat inactivated cells of different MRSA and MR-coagulase-negative strains to serve as donors. Also they constructed several two-component system (TCS) and comG and comE deletion mutants to test the impact of these genes/systems in the capacity of the recipient cells to incorporate exogenous DNA. Both planktonic and biofilm states were tested. The focus of the study – the mechanism of SCCmec transfer - is still an open and key question in methicillin-resistant *S. aureus* (MRSA) biology, with implications in infection control and treatment of infections caused by these bacteria. I congratulate the authors for the gigantic effort and the immense amount of work done to prove their hypothesis. However, the study presented is not totally convincing mainly due to methodological issues that require further clarification. Also some of the conclusions are overstating and do not completely agree with the results shown.

We thank the reviewer for encouraging comments and the following detail constructive criticism that helped us to improve the manuscript.

Major comments

i) The assays of transformation in planktonic cells and in biofilms, were done with different proportions of donor DNA vs recipient cells and for that reason efficiencies of transformation in these two situations are not directly comparable;

R1-1 The ratios of Donor:Recipient are comparable between the Biofilm and Planktonic assays. The information of the ratio was added in the revised manuscript

Line 521 (planktonic assay): The ratio of heat-killed donor (5×10^{10} c.f.u. equivalent) and recipient (after transformation assay, 1×10^{10} c.f.u.) was approximately 5.

Line 540 (biofilm assay): The ratio of heat-killed donor (2.5×10^8) and recipient (after

transformation assay, $1.5\sim 4\times 10^8$) was approximately 3.

However, the efficiency of contact between recipient and donor cells would be different in the two conditions even with the similar Donor:Recipient ratio. We avoided direct comparison and described the finding that Nef (that was not transformable in planktonic conditions) became transformable in biofilm settings. We also added this discussion in the revised manuscript and divided the original figure 2c into two Planktonic and Biofilm parts.

Line 179: Under biofilm-forming conditions, the transformation frequency in Nef reached $10^{-6\sim 7}$ at day 3 (Fig. 2c right), which remained undetectable in planktonic growth conditions (undetected, $<10^{-11}$, n=5) (Fig. 2c left).

Line 382: One possible explanation is better or more sustained cellular contact between donor and recipient cells in biofilm versus planktonic conditions even with a similar donor:recipient ratio.

ii) Transformation assays were still done in very artificial conditions, using mostly synthetic media, adding cell wall targeting antibiotics, and using a high amount of donor DNA, which is far from the real situation. It is thus difficult to convince that the phenomenon of transformation shown in the paper is in fact natural.

R1-2 We apologize for this confusion. We intended the term “natural transformation” as the opposite of “artificial transformation” that employs chemical/electroporation procedures. The term “natural transformation” was thus revised to “natural genetic transformation” that is probably more suitable to refer to the bacterial ability to acquire exogenous genetic information via the expression of competence machinery. This was explained in the Introduction and the word ‘transformation’ was corrected to ‘natural genetic transformation’ where it was necessary/ appropriate.

Line 66: Another bacterial HGT mechanism, termed “natural genetic competence/transformation,” refers to the bacterial ability to incorporate extracellular genetic information by expressing competence machinery (DNA incorporation machinery).

To answer the reviewer’s concern about dependency of synthetic media, we tested some non-synthetic media and found that milk mixed with BHI (designated as MB in revised manuscript) is suitable for staphylococcal transformation, indicating that natural transformation phenomena is not limited to synthetic media. The data was added to **Supplementary Fig. 10**.

iii) Is not clear if primers used for checking the integration of SCCmec in the recipient strains upon transformation target only the SCCmec or also the integration region in the recipient. This is crucial to understand if the SCCmec was actually integrated in the chromosome of the recipient.

R1-3 The primers used to amplify SCCmec IVa target the SCCmec alone (attL-F & attR-R) (locations were depicted in Figure 3e). The primers used to amplify SCCmec II (Xsau325 & 3.0-R) locate outside

of SCC*mec*, however, they cannot distinguish between the recipient and donor chromosome. We tried different primers and could show chromosomal integration in 9s and 35s (please see **Supplementary Fig. 15**).

Genome sequencing of transformants 35s[N315], 35s[CoNS15], 9s[N315] also confirmed the chromosomal integration (**Supplementary Fig. 16**).

iv) It is not clear how the authors dealt with the possible contamination of the recipient cells with donor DNA after mixing. For example addition of DNase after mixing donor heated inactivated cells with recipient could have avoided possible contaminations. In a PCR reaction any contamination even if in low amount would come out as a positive result. How did authors guaranteed that PCR amplification of *mecA* and SCC*mec* element regions correspond to DNA inside the recipient cells and not to contamination with the donor DNA? There are several proteins at cell surface that have DNA binding capacity, like Atl, that could have recruited donor DNA to the surface of the recipient cell.

R1-4 In order to eliminate the possibility of donor DNA contamination, we successively proliferated the transformant, which gradually eliminates donor DNA to an undetectable level by PCR. This procedure is illustrated in **Supplementary Fig. 13** in the revised manuscript.

We also confirmed that DNase and PMA does not reduce the signal (**Supplementary Fig. 14**).

Line 269: PMA-PCR and DNase treatment excluded the possibility that the *mecA* signal was from contaminating, extracellular donor DNA (**Supplementary Fig. 14**).

v) The MSSA and MRSA strains used as recipients and donors were not characterized for their genetic background. The few isolates for which the genetic background was characterized all belonged to CC5, which is the same genetic background as N315 strain. The CC5 genetic background was previously described to have a higher recombination rate and higher genetic diversity than the remaining *S. aureus* clonal complexes, and transformation ability might be a characteristic of this specific clonal lineage. It is thus difficult to evaluate how global is the phenomenon of transformation in the entire *S. aureus* population.

R1-5 CC types were determined in 99 MSSA strains following the method described by (Schwalm ND et al. 2011, ref#28) using multiplex PCR that targets virulence genes associated with the major MRSA clonal complexes. The results were added to **Supplementary Table 2**. Transformable strains 1s, 9s, 11s, 35s, and 98s were CC133, untypable, CC45, CC8, and untypable, suggesting that transformation phenomenon is not limited to CC5 (N315). We appreciate this important comment.

vi) Although transformation in *S. aureus* was achieved, apart from the expression of *comG* no clear molecular evidences were provided that a competent state was induced, that other competence genes in the genome were involved and that donor DNA interacted with the *com* genes during this process. Complementation of the results obtained with expression data, DNA-protein binding assays, and/or co-localization experiments of DNA and *com* genes using microscopy or other approaches would help to better sustain your results.

R1-6 In order to add more molecular evidence for competent state induction in the cells, we have constructed a double-reporter of P_{comG} -GFP and P_{comE} -dsRed. GFP and dsRed signals were colocalized in the same cell, indicating that, in competent cells, both *comG* and *comE* operons are being expressed:

Figure for Reviewer

Nef carrying pHY- P_{comE} dsRed- P_{comG} -gfp was grown in CS2 medium with shaking. The population percentage expressing the reporter was determined after 12-14 h of growth by fluorescent microscopy. The mean of $n = 3$ independent experiments is shown with SD. 95% of GFP positive cells were also positive in dsRed, indicating that P_{comG} activity is the suitable indicator for the competence gene expressions. No signals were detected from Δh carrying pHY- P_{comE} dsRed- P_{comG} -gfp. Scale bars, 5 μ m.

vii) The original N315 isolate, besides containing the phage Phi-N315 also contains at least one plasmid (pN315), however authors do not mention whether the strain used as donor was cured from this plasmid. This is important because plasmids can also be involved in SCCmec transfer. This observation is also valid for the other strains used as donors, including MRSA and MR-CoNS. Did the authors determine the whole genome of these strains to make sure they did not contain phages or conjugative plasmids? This is important because transduction and conjugation could have been used instead of transformation as mechanism of SCCmec transfer.

R1-7 We did not eliminate the plasmid (pN315) from our recipient. pN315 has *blaZ blaI* system (beta lactamase gene and its regulator) that plays a role in regulating *mecA* expression and accepting the SCCmec (Arede et al. 2013, ref#46). According to the reviewer's comment, we also tested the presence of *bla* in strains 1s ~ 40s and found that all of the strains confirmed to be *mecA*-transformable (1s, 9s, 11s, 35s) carried the *bla* genes (information was added to **Supplementary Table 2**). This possible necessity of *blaZ* plasmids is discussed in the revised manuscript:

Line 355: In line with a report that β -lactamase regulators affect *mecA* regulation⁴⁶, the

tested MSSA strains that could be transformed with SCCmec (1s, 9s, 11s, 35s) had plasmids and, importantly, were positive in *blaZ* and *blaI* genes usually localized to plasmids (supplementary Table 2).

Our transformable MSSA (1s, 9s, 11s, 35s) have plasmids (**Figure below**) and phage genes but lack conjugative genes (*tra/nes*) (**Supplementary Fig. 19**). Phage lysogeny is common in *Staphylococcus aureus*. Therefore, in addition to the Nef recipient (lacking phage), we also demonstrated that, in 9s, observed gene transfer is dependent on *comE* operon genes that are dispensable for phage transduction and pseudotransformation (Morikawa et al., 2012, ref#17). Thus, the observed phenomena are not conjugation, and not phage dependent transduction/ pseudotransformation.

In the present study, donor cells were heat-killed and cannot produce phage particles or conjugative machineries.

Transformable MSSA isolates (Nef, 1s, 9s, 11s, 35s) have plasmids.
Plasmid DNA was purified from log-phase cells.
RN4220 has no plasmid.
M: DNA size marker, λ *HindIII*.

viii) Another characteristic that was shown to be important for the success of the SCCmec transfer is the presence of *blaZ* and its regulators. Do the authors know if the recipient strains tested contained these genes?

R1-8 As mentioned above, we tested *blaZ* & *blaI* by PCR for 40 MSSA. 50% of screened 40 MSSA isolates have *blaZ* & *blaI* (results were added in **Supplementary Table 2**). All confirmed transformable strains were found to have these *bla* genes.

ix) Although I agree that authors provided evidences that SCCmec can be transferred by transformation, the efficiency of this process appears to be very low. When clinical strains were used as recipients only one (35s) was able to stably maintain SCCmec. I do not think that authors provided enough evidence to support that transformation is the main mechanism of SCCmec transfer.

R1-9 We screened additional 59 clinical MSSA (41s~99s) isolates using heat-killed MRSA donor (COLw/oφ:Clonal complex 8) and found more transformable MSSA isolates (**Supplementary Table 2**). We added further analysis about clinical isolate 98s to show there are multiple clinical isolates that could stably and reproducibly accommodate SCCmec. (**Table 1, Fig. 3f**). The manuscript was

modified as follows:

Line 244: We tested 20 MSSA clinical isolates using MR-CoNS8 as a donor (for 1s-20s), 20 MSSA using MR-CoNS3 as a donor (21s-40s), and 59 MSSA using COLw/oq as a donor (41s-99s) and found that approximately 20 strains of various clonal complex types were able to form colonies under cefmetazole selection (Supplementary Table 2). Among these, 5 (1s, 9s, 11s, 35s, 98s) were selected for further analysis (Table 1, Supplementary Table 5).

x) The hypothesis raised by this study is that SCCmec can be transferred by natural transformation in biofilms. This implies that emergence of MRSA occurs during infection in polymicrobial biofilms. In what particular clinical conditions do you preview this might occur? This can be included in the Discussion section.

R1-10 We added the following sentences in the Discussion.

Line 409: Mixed biofilms of *S. aureus* and other staphylococci are thus general hot spots for HGT. Although this study did not explore specific situations for MRSA emergence, polymicrobial biofilms in which CoNS and *S. aureus* coexist would be likely places to observe this phenomenon. We must note that the *mecA* gene is shared between *Staphylococcus* species from both humans and animals, similarly to other antibiotic resistance genes⁵⁹.

Based on the above finding that transformation happens in MB (milk + BHI), we assume polymicrobial biofilms in livestock environments (e.g. mastitis) might be among the candidate places where MRSA emerges through transformation, but it would be superfluous to discuss this for the time being and is thus not stated in the manuscript.

xi) Limitations of the study should be included in the Discussion section.

R1-11 We added the following sentences in discussion section.

Line 360: Although the present study did not address the factors that stabilize the SCCmec, the methodology of SCCmec transfer established here would be invaluable to detail the factors that define *mecA* stability, including involvement of DNA methylation, restriction modification systems, or CRISPR.

Line 370: however, how the biofilm structure or matrix components increase transformation efficiency remains unknown.

Line 410: Although this study did not explore specific situations for MRSA emergence,

Specific comments

Page 1, Title,

The title is misleading; MRSA has been shown to emerge by other transfer mechanisms besides transformation, like transduction and conjugation. Also although authors showed that SCCmec was transferred by transformation, this was achieved in very artificial conditions, so I wonder if “natural

transformation” is the more adequate term. Please change the title to accommodate these ideas.

R1-12 With additional evidence of 1) multiple transformable clinical MSSA isolates, 2) transformation in non-synthetic media, and 3) evidence for integration of SCC into chromosomes, we think we succeeded to demonstrate the transformation as a causative process of MRSA emergence. While phage transduction can transmit small SCC_{mec} types, it’s unable to do so for large SCC and, importantly, the involvement of *ccr* has not been shown. Nonetheless, as alternative mechanisms, phage transduction or pseudotransformation may play roles which have been carefully described in the manuscript.

As mentioned above, we intended that the term ‘Natural transformation’ represents bacterial ability to transform by incorporating DNA, and not ‘natural’ settings where transformation occurs. In order to clarify this, the term ‘natural genetic transformation’ (more often used as the term to distinguish from artificial transformation) was used in the revised title.

Page 2, Introduction, line 43-45:

Please verify if MRSA is the leading cause of nosocomial infection. Antibiotic-resistant Gram-negative bacteria nosocomial infections are increasing and are now a major concern. In the community CA-MRSA is also decreasing and LA-MRSA is not very frequent in human infection.

R1-13 As the reviewer pointed out, Gram negatives are increasing, while MRSA is still the leading cause according to a 2019 CDC report. MRSA still remains as a big burden and now it is recognized as an ESKAPE pathogen together with other AMR bacteria. In the revised manuscript, ‘leading cause’ was rephrased as ‘major cause’, to avoid the wrong impression that MRSA alone is the problem.

reference: CDC report 2019

<https://www.cdc.gov/drugresistance/pdf/threats-report/2019-ar-threats-report-508.pdf>

MRSA: 323,700 cases & 10,600 deaths

ESBL-producing Enterobacteriaceae (Gram negative) (including E. coli, Salmonella etc):
197,400 cases & 9,100 deaths

Page 2, Introduction, line 46:

Specify the type of percentage you are talking about. Is it of nosocomial infection? Or other?

R1-14 It refers to the percentage of MRSA among *S. aureus* isolates from inpatients as reported in the Center For Disease Dynamics, Economics & Policy. The manuscript was revised as follows:

Line 45: The percentage of MRSA among *S. aureus* isolates from inpatients differs between countries (Vietnam 73%, United States 45%, Japan 41%, North Europe 1%)⁶,

Page 2, Introduction, line 53-55

Methicillin resistant determinant was found outside SCC_{mec} in some coagulase-negative staphylococci, like *S. fleurettii* and some *S. vitulinus*. In these species *mecA* it is located in the chromosome, outside any mobile genetic element. Please rephrase.

R1-15 Thank you very much for this information. We understand that reviewer means *mec* homologues can be found in the chromosome. We corrected the description.

Line 52: In MRSA, the methicillin-resistant determinant *mecA* is always located within the SCC (*SCCmec*) while its homologues can be found in SCC, chromosomes, or plasmids in *Staphylococcus* or *Micrococcus* species¹⁰.

Page 2, Introduction, line 56

Indicate what do CcR stands for. Also the function of *orfX* was already described and the name of the gene was changed. Please update this information.

R1-16 Rephrased as follows:

Line 55: *SCCmec* is itself a 20-60 kb genetic element integrated by Ccr (cassette chromosome recombinases) at a specific site (attachment site, *attB*) in *orfX* (a.k.a. *rmlH*, encoding rRNA methyltransferase¹¹) near the replication origin of the chromosome¹².

Page 3, Introduction, line 68-70

Add that transformation was achieved only in the presence of a particular growth medium (CS2).

R1-17 We included the medium dependency as follows:

Line 72: In these cases, bacteria (N315 derivative strains) that are modified to overexpress SigH incorporated *SCCmec* II elements from purified genomic DNA in a manner dependent on a particular growth medium (CS2).

Page 3, Introduction, line 77

Add information on other studies wherein *SCCmec* transfer was accomplished using other mechanisms such as transduction and conjugation.

R1-18 Transduction and conjugation were described in the Introduction:

Line 59: Although short or fragmented *SCCmec* can be transmitted by transduction¹⁴ or conjugation¹⁵,

Page 3, Results, line 78

It is not obvious why did the authors looked in the first place into TCS in the context of natural transformation in *S. aureus*. This has to be contextualized. Has this association been done before in other bacteria? Why were deletion mutants constructed for this type of system in particular? Some information is missing here.

R1-19 We have now revised and adjusted the text in the introduction section as follows:

Line 82: Previous analysis showed that *S. aureus* usually does not activate the competence operon promoter (*P_{comG}*) but, when cultivated in CS2 medium,

subpopulations expressing P_{comG} reporter increase up to ~10%¹⁷. This suggests that certain environmental or intrinsic cues are necessary for natural transformation.

Page 4, line 95

How do authors know that the Nef strain does not contain any conjugative element or lysogenic phage?

How did authors deal with genes with unknown function in this strain?

R1-20 N315 genome sequence does not contain any known conjugative elements. Single phage in N315 was eliminated to obtain the strain Nef.

We also confirmed the absence of phage and conjugative genes in Nef by PCR as mentioned above (**Supplementary Fig. 19**). In addition, to ensure that the observed phenomenon is natural genetic transformation rather than phage-dependent phenomena or conjugation, we included Nef $\Delta comE$ (deletion of *comE* operon in Nef) as a negative control in all of the transformation assays.

Page 4, line 99

Can you please provide the composition of the CS2 medium? This is important to understand if any component in particular can be inducing the competent state. What this medium has that is so crucial for the occurrence of transformation? In the Lab, when preparing competent cells it is important to add CaCl₂ so that DNA binds to cell surface. Is there any component doing the same function in CS2 medium? If this is the case I do not think that authors can say this is a natural phenomenon.

R1-21 CS2 composition was added as a **supplementary Table 7**.

It is known that Ca²⁺ is essential for natural genetic transformation in other bacteria such as *Streptococcus pneumoniae*, and CS2 contains ~30 μ M Ca²⁺.

Page 4, line 104

I do not think that growth defects is the correct term to use, since authors did not look into cell morphology. What authors are probably observing is cell lysis in the stationary phase. Bacteria grow and divide mainly in the exponential phase what is not frequent in the stationary phase.

Page 4, line 105

Please indicate to what specific yield are you referring to. Do you mean higher OD?

R1-22 We have addressed the above two points together. The manuscript was improved as follows:

Line 110: The Δ TCS strains did not exhibit an altered growth curve (O.D._{.600}) in TSB (Supplementary Fig. 2a). However, in CS2 medium, Δ 5, Δ 12, and Δ 13 exhibited a minor reduction in O.D._{.600} around 8~12 h compared with Nef while Δ 9 and Δ 17 exhibited a higher O.D._{.600} at the stationary phase (Supplementary Fig. 2b).

Page 5, line 113-114

Authors refer in the text that complementation in trans within these delta-TCSs mutants restored the percentages of GFP-positive cells however they do not provide the figures showing this. Please include

microscopy images corresponding to the % of GFP-positive cells for the complemented strains.

R1-23 Following the reviewer's advice, we have now included the microscopy images in **Figure 1c**.

Page 5, line 116

With the results showed up until this point the authors can only affirm that TCS12, TCS13, and TCS17 are involved in the regulation of the *comG* not the entire competence machinery. Competence is a cell state provided by a plethora of genes not the expression of a single gene. Actually later in the paper authors reach to the conclusion that *comG* is not essential for natural transformation, which is somehow contradictory.

R1-24 SigH is the master regulator of competence in *S. aureus* that directly controls the expression of *P_{comG}* and *P_{comE}* competence operons. We selected the *P_{comG}-gfp* reporter as one of the indicators of SigH-expressing cells. As described above, we constructed a dual-reporter of *P_{comG}-GFP* and *P_{comE}-dsRed*. GFP and dsRed signals were colocalized in the same cell, indicating that in competent cells both *comG* and *comE* operon genes are being expressed.

Page 5, line 121-122

What were the breakpoints used by the authors to consider a strain as being resistant or susceptible to vancomycin, bacitracin and nisin? Clinical breakpoints are not defined for TSB or CS2 medium.

R1-25 Vancomycin MIC break points: (S) ≤ 2 , (I) 4~8, (R) ≥ 16 $\mu\text{g mL}^{-1}$

Bacitracin, Nisin: Breakpoints are not set in CLSI standard.

Among the three antibiotics, only vancomycin has an official susceptibility breakpoint in *S. aureus* defined by CLSI. We have now added the MIC values of vancomycin in MH medium in **Supplementary Table 4**:

In addition, we have rephrased "Indeed, we confirmed that $\Delta 12$ is susceptible to vancomycin and $\Delta 17$ is susceptible to bacitracin and nisin in CS2 and TSB media" to:

Line 141: Indeed, we confirmed that $\Delta 12$ is more susceptible to vancomycin and $\Delta 17$ is more susceptible to bacitracin and nisin than Nef (Supplementary Table 4).

Page 5, line 134-137

When treating cells with vancomycin and bacitracin the bacterial growth may be affected as well as global gene expression. Could it be that the decrease in expression is not specific of *comG*, but could be a global decrease in gene expression due to a decrease in growth? This could be overcome if the expression of a control gene not involved in competence would have been used or by showing that the growth rate did not change in the presence of antibiotics.

R1-26 Following this suggestion, we have now added the OD graphs of each group to **Supplementary Fig. 5**.

Page 6, line 136-137

I am not sure that authors can affirm that there was an increase in comG-GFP reporter expression when cells were exposed to nisin. Is the increase observed at 8h statistically significantly different from that observed at 4 ug/mL? If the difference is not significant then nisin has an effect similar to that of vancomycin and bacitracin, decreasing the comG-GFP expression with the increase in the concentration.

R1-27 No statistical significance was found, that is why we used the word “slightly”. We have now added further clarification to the text:

Line 156: On the other hand, 8 or 16 $\mu\text{g mL}^{-1}$ nisin slightly, but reproducibly, increased reporter expression intensity in Nef-GFP (Supplementary Fig. 5c), though not in a statistically significant manner.

Page 6, line 143

Authors say that the percentage of GFP-expressing cells in biofilm was less than in planktonic culture, and that data indicates that the Agr quorum sensing TCS13 plays a role in comG reporter expression. However, they do not specify what data are they taking about. Please specify.

R1-28 We apologize for this confusing statement. The description was modified to simply state that %GFP expressing cells are higher in planktonic growth (Fig. 1c: ~10%) than in biofilm (Fig. 2a: ~1%), and moved to the Discussion section.

Line 379: the P_{comG} reporter expression did not increase in biofilm compared with planktonic condition (Fig. 2a: ~1% vs Fig. 1c: ~10%)

Page 6, line 156

Please clarify if tetracycline resistance gene in the donor strain was encoded in the chromosome or in a plasmid and what was the size of this plasmid.

R1-29 Tetracycline resistance gene in these donors is in the chromosome.

Line 174: Nisin and biofilm conditions were tested for their effects on transformation efficiencies by using heat-killed donor cells carrying a chromosomal tetracycline resistance gene (N315 Δ cls2-tet^R, or Nef Δ cls2-tet^R).

Page 6, line 158

Authors showed that in Nefh, the transformation frequency was similar to Nef under biofilm-forming conditions. These results and results obtained with clinical strains suggest that in biofilms transformation might be independent on sigH. This would represent a totally new mechanism different from the one described previously by the authors. This should be clear in the paper.

R1-30 We confirmed that sigH-null strain (Nef Δ h) is not transformable in the biofilm condition. This was included as **Figure 2e** in revised manuscript.

Page 7 line 167-169

Unexpectedly, authors found that the negative control (deletion mutant for *comG*) was transformable, however, all the experiments performed previously used this gene as a reporter for the occurrence of transformation. How do the authors conceal the previous results with this observation?

R1-31 As mentioned above, *PcomG* is a good representative promoter to monitor SigH activity, and *PcomG* positive cells correlates with *PcomE* positive cells.

Page 8, line 189-190

The influence of genetic background in the ability of strains to stably maintain the SCCmec was previously reported. The knowledge on the genetic background of transformable and non-transformable strains should be provided, because this would indicate if the phenomenon of transformation described here is specific of certain genetic backgrounds and conditions or a more global occurrence in *S. aureus*.

R1-32 We checked the clonal complex (genetic background) of the transformable and non-transformable strains (information was added to **Supplementary Table 2**) and found that is quite diverse, even among the transformable strains, indicating that transformation phenomenon is not limited to one specific genetic background / clonal complex. Thank you again for this comment.

Page 8, line 197-198

It is possible that SCC elements are transferred through non-conjugative plasmids if conjugative plasmids are also present in the same strains or if the plasmid is transferred within a phage. A SCC element containing *mecB* was already found in *S. aureus* in a plasmid, this has also been described in *Micrococcus caseolyticus*, a genera very closely related to *Staphylococcus*, which appear to be involved in the emergence and evolution of SCCmec. Please rephrase to include this information.

R1-33 Manuscript has been corrected and our scope on *SCCmecA* was clarified.

Line 52: In MRSA, the methicillin-resistant determinant *mecA* is always located within the SCC (*SCCmec*) while its homologues can be found in SCC, chromosomes, or plasmids in *Staphylococcus* or *Micrococcus* species¹⁰.

Line 236: to the best of our knowledge, *mecA* has not been found in such a plasmid-carrying SCC in staphylococcal isolates. In this study, we propose that natural transformation in biofilms is the major mechanism for cell-to-cell transfer of *SCCmecA* based on the following experimental evidence.

Page 8, line 204

More information should be provided on the 20 MSSA strains used as recipients. To what genetic background do they belong? This is important information because not all the isolates were transformable and the genetic background was reported to have a role on genetic transfer of SCCmec.

R1-34 As described above (**Supplementary Table 2.**)

Page 10, line 235-236

The impact of *attB* (chromosome) and *attS* (SCC*mec*) sequences and adjacent sequences in the integration of SCC*mec* by Ccr has been previously reported (Wang et al. 2012) and could eventually explain the lack of stability of SCC*mec* in the recipient strains. Have the authors checked the diversity in the *attB* and *attS* sequence of recipient strains and SCC*mec*? Differences in the efficiency of integration in the chromosome can also derive from difference in methylation patterns and R/M systems or the presence of CRISPR in the recipient strains? The results obtained regarding stability should be discussed in these contexts.

R1-35 We have sequenced the *attB* region in transformable MSSA (1s, 9s, 11s, 35s) and nontransformable MSSA isolates (2s, 17s, 20s, 40s) (**Supplementary Fig. 18**). The 15 bp of the core *attB* sequence is conserved among tested strains except the nontransformable 20s. The downstream regions, which were previously reported to affect integration efficiency (Wang et al. 2012, ref#47), were variable among tested strains. This difference may affect the frequency of SCC*mec* integration or stability. These were added to the Discussion.

Line 358: The sequences encompassing the *attB* site are also known to affect the efficiency of chromosomal SCC integration⁴⁷ and our tested MSSA strains were diverse in the *attB* downstream sequences (Supplementary Fig. 18). Although the present study did not address the factors that stabilize the SCC*mec*, the methodology of SCC*mec* transfer established here would be invaluable to detail the factors that define *mecA* stability, including involvement of DNA methylation, restriction modification systems, or CRISPR.

Page 10, line 236

Authors should clarify what kind of primers did they use (targeting what) to detect the presence of SCC*mec* in the recipient strain upon transformation. The PCR should have been designed to amplify the junction site between recipients DNA and the SCC*mec* and sequencing done to confirm. If the PCR reaction only targeted the *mecA* or fragments of the SCC*mec*, amplification could correspond to contamination of the recipient DNA with donor DNA.

R1-36 As described above, chromosomal integration of SCC was confirmed by appropriate primers (**Supplementary Fig. 15**) and genome sequencing (**Supplementary Fig. 16**).

Page 11, line 263

Please clarify which was the genetic determinant that was transformed into the recipient strain when transformation was tested in biofilms in the presence of the cell wall-targeting antibiotics.

R1-37 Donor and other information were added in the revised **Fig. 2d**.

Page 12, line 287

I think authors meant SCCmec I-IV instead of SCCmec I-V?

R1-38 We apologize for this mistake and corrected it on **Line 322**.

Page 13, line 311

Proposing that natural transformation in biofilms is the main means of SCCmec transfer is overstating. The conditions described in this study are somehow artificial (CS2 medium) and assume that in the biofilm total DNA of the donor is available. Moreover, there is no direct comparison with other transfer mechanisms.

R1-39 As described so far in this letter, we think we could add enough evidence supporting that the natural genetic transformation is the mechanism that can transfer SCC, especially 1) natural genetic transformation is the only known way that can transfer long type II SCC, 2) involvement of *ccr-attB* is shown only for the natural genetic transformation. We hope the latter point would be tested in other HGT systems in future studies by our scientific community. The description was modified.

Line 344: To the best of our knowledge, this study is the first to show that intercellular SCC transmission requires the *ccr-attB* system and, based on this evidence, we propose natural transformation **as the route for SCCmec transmission**.

Page 13, line 312

The transformation efficiency occurring in the natural environment should be largely decreased when compared to the laboratory assays of this study, because the growth media will not be CS2, the concentration of DNA of donor would be lower and compatibility of donor and recipient DNA must exist so that SCCmec is acquired and maintained.

R1-40 We added an example where CS2 is not required (MB: **Supplementary Fig. 10**).

Page 13, line 319-321

Authors cannot affirm that they tested the impact of genetic background on the efficiency of transformation because they did not characterize the genetic backgrounds of the donors and recipient strains.

R1-41 Clonal complex and *bla* presence were tested and included in **Supplementary table 2**.

Page 13, line 323-324

In my opinion the assays performed do not allow for a direct comparison of the efficiencies of transformation between the planktonic and biofilm conditions because the conditions in planktonic and biofilm were not exactly the same. In particular the number of recipient cells and amount of donor

DNA were not normalized in the two assays.

R1-42 Direct answer was described above (R1-1), and we also modified the text:

Line 365: The present study establishes a reliable experimental system to detect natural transformation in *S. aureus*.

Page 14, line 333-335

This sentence is not clear. Please rephrase.

R1-43 The sentence was clarified as follows:

Line 378: Moreover, it is also likely that the transformation process following competence gene expression is facilitated in biofilms because i) the P_{comG} reporter expression did not increase in biofilm compared with planktonic condition (Fig. 2a: ~1% vs Fig. 1c: ~10%) but transformation was efficient in biofilms and ii) artificial SigH-overexpression did not increase transformation efficiency in biofilms (Fig. 2c).

Page 14, line 339-342

Please elaborate more on the relation between nuclease heat inactivated cells and transformation occurring in biofilms. This is not very clear.

R1-44 Explanation was added as follows:

Line 387: it was crucial to use heat-killed donor cells (cellular DNA resides within a peptidoglycan cell wall that cannot be disrupted by heat) rather than purified chromosomal DNA.

Page 14, line 343-344

Please explain better the relation between eDNA sequestration in biofilms and the fact that genomic purified DNA cannot serve as a transformation donor but heated inactivated cells can. Do authors think that donor proteins or metabolites might be important to induce the competent state in the recipients?

R1-45 Thank you for this comment. We assumed that DNA is protected from nuclease in the cell wall peptidoglycan of the heat-killed cell and remains available until the cell wall is degraded by the recipient autolytic enzyme. But the Reviewer's comment is also a possibility. We added these possibilities for readers to be able to deduce the reason.

Line 394: It is also elusive whether certain donor factors facilitate transformation or certain recipient activities (e.g., peptidoglycan hydrolases) cooperatively work to retrieve the DNA packed and protected in the cell wall.

Page 15, lines 357-358

It is not proven that in colonization state *S. aureus* form biofilms. Usually the cfu load is low during skin or mucous membranes colonization. In fact Krismer et al (2011) point towards a dispersed rather

than a biofilm-related mode of growth during *S. aureus* nasal colonization.

R1-46 Thank you for the correction. The description was simplified:

Line409: Mixed biofilms of *S. aureus* and other staphylococci ~~formed during commensal state or co-infections~~ are thus general hot spots for HGT.

Page 19, lines 461

Do $5 \log_{10}$ stand for cells or for CFU/mL?

R1-47 It is cells (CFU equivalent quantity). Clarified as below:

Line 521: heat-killed donor (5×10^{10} c.f.u. equivalent)

Page 32, Figure 3 Legend

Figures are of very poor quality. Please try to produce ones with better resolution in which the bands are more visible, including those of the marker.

R1-48 The quality of electrophoresis was improved. (**Figure 3**)

Supplementary Table 4

Add information about the units (CFU/mL?) and what it means the number in parenthesis.

R1-49 Information added.

Table1 note:

The mean of **Transformation frequencies** (number of transformants/ c.f.u of recipient) is shown with \pm s.d.

n: number of independent experiments.

ND: none detected (c.a. $< 10^{-9}$)

General comment: In the description of the transformation experiments, including the Figure legends, it is almost always missing the strain that was used as donor. Also it is missing the information if heat-killed cells or purified genomic DNA was used. This is important because the donor and type of DNA can also have an impact in the outcome of transformation. Methylation pattern and R/M systems might have a role. Please go through the text and add this information where is missing.

R1-50 We apologize for this insufficient presentation. We have included information about transformation experiment in each Figures and Supplementary Figures.

Reviewer #2 (Remarks to the Author):

The manuscript by Maree et al., is dedicated to investigate natural transformation in *Staphylococcus aureus*. The study is divided in two parts. In the first part the authors investigated how the presence of antibiotics, two-component systems and biofilm lifestyle affect the activity of sigH transcription factor. In the second part, the authors explored the transferability of the SCCmec element from heat-killed bacteria to different recipients in biofilm conditions.

The biological question that the authors are trying to address is important and the manuscript lists several interesting observations related to the capacity of *S. aureus* to gain foreign DNA.

We appreciate the reviewer for the positive comments and the following interesting concerns about roles of TCS systems.

Major comments

- Three two-component systems (AgrCA, BraSR, and VraSR) out of the sixteen present in the genome of *S. aureus* were identified as important for the activity of SigH. AgrCA and BraS induce whereas VraSR represses the activity of sigH. Because the regulon of each of these TCS includes dozens of genes, it would be important to know which is the contribution of the modification of SigH expression to the DNA uptake phenotype of AgrCA and BraSR mutants. Can constitutive overexpression of SigH complement the DNA uptake phenotype of BraSR and AgrCA mutants?

R2-1 We addressed these points. In brief, expression of *sigH* could restore the deletion effect of these TCSs. Results were included as **Supplementary Fig. 3, Supplementary Fig. 9** and in the manuscript as follows:

Line 128: We then introduced the pRIT-sigH^{NH7} plasmid, which allows constitutive transcription of *sigH* mRNA with an intact 5'UTR¹⁷ into $\Delta 13$ and $\Delta 17$, together with the P_{comG} -*gfp* reporter, generating strains $\Delta 13$ -NH7-GFP and $\Delta 17$ -NH7-GFP. Of note, the 5'UTR has inverted repeat sequences and is thought to suppress SigH translation (Supplementary Figure 3a). In TSB medium, GFP expression was not observed in $\Delta 13$ -NH7-GFP, $\Delta 17$ -GFP-NH7, or Nef-NH7-GFP, indicating that translational suppression is sustained. In CS2 medium, introduction of the pRIT-sigH^{NH7} plasmid restored GFP expression in both the $\Delta 13$ and $\Delta 17$ TCS mutants (Supplementary Fig. 3b). Thus, the absence of TCS13 and TCS17 can be compensated for by the overexpression of *sigH* mRNA as measured by P_{comG} -*gfp* expression.

Line 200: In addition, overexpression of *sigH* by pRIT-sigH^{NH7} or by pRIT-sigH (with a modified 5'-UTR to release the translational suppression of *sigH*) restored biofilm formation and transformation frequencies in $\Delta 13$ and $\Delta 17$ (Supplementary Fig. 9).

- The results showed that DNA uptake occurs more efficient in biofilms than in planktonic cells. At the same time the authors found that mutants in AgrCA and BraSR displayed a reduced capacity to form biofilm. However, the authors have not explored how biofilm formation and DNA uptake phenotypes are related with BraSR, AgrCA activity. One possibility to explore this likely connection will be to identify a medium (TSB, BHI, M9) where BraSR and AgrCA mutants retained the capacity to produce biofilm. Then, the authors could explore the efficiency of DNA uptake of BraSR and AgrCA mutants in biofilms.

R2-2 We appreciate the reviewer for this thoughtful comment. We have now found that cow milk mixed with BHI (MB) results in strong biofilm formation even in Δ TCSs. Transformation was detectable in this condition, and especially TCS17 (BraSR) mutant had a notable increase in the transformation suggesting that transformation efficiency is mediated *via* biofilm formation. The data was included as **Supplementary Fig. 11**.

Line 210: To gain insight into how biofilm formation and transformation are related to TCS13 and TCS17 functions, we tested the transformation of Δ 13 and Δ 17 in MB where we found both mutants could make rigid biofilms (Supplementary Fig. 11a). The transformation frequencies of Δ 13 and Δ 17 were comparable to Nef in MB (Supplementary Fig. 11b), making it likely that TCS13 and TCS17 indirectly contribute to transformation *via* biofilm formation in CS2 but are dispensable in MB.

- The presence of subinhibitory concentration of vancomycin or bacitracin affect the activity of SigH. The effect of these antibiotics on cell wall is pleiotropic and they might not be any connection between these observations and BraSR and AgrCA mutants deficiency in DNA uptake and biofilm formation.

R2-3 We agree with this opinion and the manuscript was modified in order to add this possibility. (Since these experiments are about tetracycline resistance transformation, original Figure 4 was moved to Figure 1d in revised version.)

Line 190: The effect of cell-surface affecting antibiotics is pleiotropic and, while their effects on P_{comG} activity/ transformation might be indirect, our data indicate that they may offer some degree of control over natural transformation efficiency.

- The results showing that the SCCmec element can be transferred from heat-killed bacteria to different recipients are very interesting. I assume that BraSR and AgrCA mutants were not used to investigate the capacity to accept the SCCmec element because they are unable to form biofilm in CS2 medium. Again, a medium where BraSR and AgrCA mutants retain the capacity to form biofilm could allow to test their role in SCCmec transfer. Alternatively, the capacity to produce biofilm *in vitro* is very different among *S. aureus* strains. Thus, it is very likely that mutation of BraSR and AgrCA does not affect the capacity to produce biofilm in CS2 medium in other *S. aureus* strains.

R2-4 By using the MB medium that effectively induces biofilm formation, transfer of SCCmec was observed in $\Delta 13$ and $\Delta 17$, indicating that TCS13 and TCS17 are not essential for the SCCmec transfer itself. These new data were included as **Supplementary Fig. 11c**.

Line 368: Notably, MB medium that facilitates biofilm formation allowed *tet* and *mecA* transformation in TCS13 and TCS17 mutants (Supplementary Fig. 11b,c). This indicates that these signaling systems are dispensable once biofilm forms;

- In relation with the previous question, *S. aureus* is able to produce a biofilm matrix made of proteins or exopolysaccharides. Does the nature of the biofilm matrix affect the transformation efficiency?

R2-5 The present study did not clarify this point, but we tested 1) *sarA* mutant of Nef and 2) effect of DNase (**Figure attached next page**).

SarA regulates *bap* and *ica* genes that encode biofilm associated protein and PIA components respectively (Trotonda et al. 2005; Valle et al. 2003). In line with this, our *sarA* mutants failed to make biofilm, and transformation was abolished, implying the importance of biofilm matrix. We also tested the effect of DNase on the biofilm and transformation, and found that DNase can abolish both. DNA is one of the components in biofilm, (but high concentrations of DNase may be degrading donor DNA too). Thus, biofilm matrix and/or its components seems to be critical for transformation. The *bap* gene does not exist in any of transformable strains (N315, 9s, 35s) whose genome sequence is available, while *ica* operon is shared among these three. Thus, *ica* operon maybe important for transformation, and it is an interesting future study. We just mentioned this very briefly in discussion as follows:

Line 370: however, how the biofilm structure or matrix components increase transformation efficiency remains unknown.

References:

- Trotonda MP, Manna AC, Cheung AL, Lasa I, Penadés JR. SarA positively controls bap-dependent biofilm formation in Staphylococcus aureus. J Bacteriol 187(16), 5790-5798.2005.
- Valle J, Toledo-Arana A, Berasain C, et al. SarA and not sigmaB is essential for biofilm development by Staphylococcus aureus. Mol Microbiol 48(4), 1075-1087. 2003.

Figure for Reviewer

DNase I treatment and *sarA* deletion impair biofilm formation and natural genetic transformation.

a (top), Nef was statically grown in CS2 medium for 3 days with different concentrations of DNase I. Transformation frequencies and CFU mL⁻¹ were determined after 3 days. A heat-killed NefΔcls2-tet^R donor was used. Transformants were selected by tetracycline. The mean of n = 2-3 independent experiments is shown with SD. **a (bottom)**, Biofilm formation of Nef was assessed in 96-well plates after static growth for 24 h with different concentrations of DNase I. The mean of n = 3 independent experiments is shown with SD.

b (top), Nef and its derivative *sarA* deletion mutant (ΔsarA) were statically grown in CS2 medium for 3 days. Transformation frequencies and CFU mL⁻¹ were determined after 3 days. A heat-killed NefΔcls2-tet^R donor was used. Transformants were selected by tetracycline. The mean of n = 3 independent experiments is shown with SD. **b (bottom)**, Biofilm formation of Nef and ΔsarA was assessed in 96-well plates after static growth for 24 h with different concentrations of DNase I. The mean of n = 2 independent experiments is shown with SD.

Minor comments

The strain NefH is not described until line 158, whereas it is used in all the panels of Figure 2 that it includes experiments described before. Thus, either the strain is described earlier in the text or it is described in the legend of Fig. 2.

R2-6 The strain Nefh has now been described in the legend of Fig. 2 as follows:

Figure 2 caption: a, The percentage of Nef and its derivatives, Nefh (overexpressing SigH), Δ13, Δ17, and Δh (*sigH*-null mutant), expressing P_{comG}-gfp reporter.

Reviewer #3 (Remarks to the Author):

The manuscript describes conditions believe to be natural transformation and builds of previous work showing that *S. aureus* can be transformable under special conditions where the *com* locus is expressed.

We thank the reviewer for the following constructive criticism that helped us to improve our manuscript. Now, we believe that the explanations for experimental conditions are improved and a series of control experiments further supported our conclusion.

There are several major concerns with this manuscript.

1) In description of the central transformation experiments it is not indicated which DNA is being used in the experiment; at which antibiotic concentration were the transformants selected and on which agar plates. In some cases, it is also unclear which recipient strain was being used. In Materials and Methods it appears that at least in some cases the transformants are incorporated in BHI agar - why is performed and why BHI?

R3-1 Information of transformation experiments (including donor, recipient, growth conditions) have been embedded in each Figure and Supplementary Figure. Selection procedures of transformants were also illustrated in the revised **Supplementary Fig. 7** (tetracycline) and **Supplementary Fig. 13** (cefmetazole). BHI is one of many complex rich media ordinarily used in our lab and there is no specific reason; TSB or others may also work. This was mentioned in Methods section:

Line 538: In this case, we chose BHI as a complex, rich medium although other media may be viable for this purpose.

2) The transformants are not checked thoroughly. It is not confirmed whether the transformants are spontaneous resistant mutants arising or whether the actual DNA being used for the transformation is present in the recipient strain background. The transformants have to be genome sequenced to demonstrate that they actually is carrying the transformed DNA and that they have not picked up any phages in the process - see below

R3-2 The presence of donor DNA (*tet*, *mecA*, or SCC) was confirmed by PCR (*tet*: **Supplementary Fig. 7**, *mec*: **Fig. 3a**, **Supplementary Fig. 14**, SCC: **Fig. 3e,f,g**). PCR (**Supplementary Fig. 15**) and genome sequencing (**Supplementary Fig. 16**) confirmed presence of SCC*mec* in the chromosome backbone.

We also added a series of negative control experiments (listed below) confirming our experimental setting never detect spontaneous resistant mutant.

Fig. 2e: Δh , Nef Δ comE

Table 1, blue tone's cells: e.g. Nef (as donor that has no *mecA*), Nef Δ comE, 9s Δ comE

Supplementary Fig. 7b: Nef (as donor with no tetracycline resistance gene)

Supplementary Fig. 11: Nef Δ comE

Supplementary Fig. 12: Nef (as donor with no tetracycline resistance gene)

3) It is not clear what the Nef-delta-cls2-tetR strain is being used for but it has been made by transduction and we know that in transduction the majority of the transductants are carrying the transducing phage (commonly phi11 or 80a) in addition to phages from the donor strain

R3-3 We were careful about this point and we confirmed that mitomycin induction does not produce any infective phage particle from Nef-delta-cls2-tetR (plaque forming unit in culture sup after the induction was 0). We also confirmed that Nef-delta-cls2-tetR has no typical phage by multiplex PCR (**Supplementary Fig. 19**). In addition, donor cells were all heat-killed (boiled) in the present study and infective phage particles are expected to denature and not be produced from heat-killed cells. Moreover, the negative control strain (Nef Δ comE) was unable to be transformed.

4) the transformation efficacy is reported as frequencies (for example supplementary table 4) but the actual number of colonies on the agar plates should be reported,

R3-4 We have now included an additional table showing the actual number of colonies counted as transformants (**Supplementary Table 5**). (Original Supplementary Table 4 was moved to main Table 1).

5) In figure 2 transformation with control DNA is missing not carrying the resistance genes that is selected for?

R3-5 We have now included the control DNA donor (heat-killed Nef that has no resistance gene) (**Supplementary Fig. 7b**)

6) In the critical transformation experiments of the SCCMec cassette there is also a need for genome sequence analysis rather than PCR analysis of transformants in order demonstrate the transfer.

R3-6 We have now genome sequenced the SCCmec transformants and confirmed the presence of integrated SCCmec in the recipient chromosomes (**Supplementary Fig. 16**), consistent with the PCR tests (**Fig. 3e,g; Supplementary Fig. 17**).

In general, the authors are doing very little to control for a number of artefacts that are obvious but will have to be accounted for particularly with the controversial results communicated. Further the authors are using a lot of space to monitor expression of comG under different conditions (all of figure 1 and figure 2a and b) but subsequently discard comG as being required for transformation (figure 2g).

R3-7 A series of control experiments have been added to the revised version as described above.

To confirm that $P_{comG-gfp}$ is the appropriate indicator for the competence genes expression, we

constructed a dual-reporter of P_{comG} -GFP and P_{comE} -dsRed. GFP and dsRed signals were colocalized in the same cell (95%), indicating that in P_{comG} -GFP active cells, a series of competence genes are being expressed.

Figure for Reviewer

Nef carrying pHY- P_{comE} -dsRed- P_{comG} -gfp was grown in CS2 medium with shaking. The population percentage expressing the reporter was determined after 12-14 h of growth by fluorescent microscopy. The mean of $n = 3$ independent experiments is shown with SD. 95% of GFP positive cells were also positive in dsRed, indicating that P_{comG} activity is the suitable indicator for the competence gene expressions. No signals were detected from Δh carrying pHY- P_{comE} -dsRed- P_{comG} -gfp. Scale bars, 5 μm .

Line 57-59: This is wrongly stated as the cited publication inferred the number of transfer events based on modelling

R3-8 Thank you for the correction. The manuscript was revised as follows:

Line 58: Evolutionary models infer that at least 20 independent acquisitions of SCCmec have occurred in *S. aureus*¹³.

line 92: what do you mean by delta3-delta17? if you refer to the mutants why is nr. 2 missing?

R3-9 We rephrased the text as follows. TCS2 is within SCC, but Nef lacks SCC.

Line96: To delineate conditions conducive to natural transformation, we generated a series of 15 TCS deletion mutants, removing each set of TCS genes (TCS3~TCS17) except the essential TCS1 (WalKR)²¹, in the *S. aureus* strain N315ex w/o ϕ ¹⁷ (termed Nef; Supplementary Table 1) (Fig. 1a). Nef is an N315 derivative strain and does not possess any conjugative elements or a lysogenic phage that transfers DNA by

transduction or pseudo-competence. **Nef also lacks the SCCmec and its embedded TCS2 (SA0066-SA0067).** The resulting Δ TCS strains in Nef background were designated Δ 3- Δ 17.

line 99: What characterizes the CS2 medium?

R3-10 The composition of CS2 medium was added as **Supplementary Table 7**. CS2 was found to be an appropriate medium to detect transformation in our previous study. This was added in the Introduction section:

Line 72: In these cases, bacteria (N315 derivative strains) that are modified to overexpress SigH incorporated SCCmec II elements from purified genomic DNA **in a manner dependent on a particular growth medium (CS2).**

figure 1c: How many cells were counted in each experiment?

R3-11 At least 300 cells were counted in each experiment. We added this to the Figure 1 caption.

line 141: Not clear where this is going and only comment in day 2 and 3 and when decrease on day 4

R3-12 Rephrased:

Line 161: In the Nef-GFP reporter strain, the percentage of GFP-expressing cells increased **and peaked at day 3** (Fig. 2a),

line 118-152: Is mostly based on supplementary information and seems very extensive to demonstrate very few points - can be shortened substantially

R3-13 According to another reviewer's comment, we included further results in terms of TCS17 and TCS13 (**Supplementary Figs. 3, 9**), which further reinforced the importance of biofilm conditions rather than antibiotic stress. In this context, we hope reviewer could agree to maintain this description. We also slightly shortened it.

line 155: The logic is not clear as you in line 142-143 state that you have less comG expression in biofilms

R3-14 We apologize for our unclear description. The description was shortened and moved to the Discussion section. The GFP reporter in planktonic (Fig. 1c: ~10%) was higher than biofilm conditions (Fig. 2a: ~1%).

Line 379: the P_{comG} reporter expression did not increase in biofilm compared with planktonic condition (Fig. 2a: ~1% vs Fig. 1c: ~10%)

line 156: what is the delta *cls2* and why is that introduced here?

R3-15 The strains have tetracycline resistance gene in chromosome (*cls2* locus). It was constructed for other purpose, and not necessarily in *cls2* locus for the present study. The manuscript was revised

as follows:

Line 174: Nisin and biofilm conditions were tested for their effects on transformation efficiencies by using heat-killed donor cells carrying a chromosomal tetracycline resistance gene (N315 Δ cls2-tet^R, or Nef Δ cls2-tet^R).

Line 509 (Method section): For heat-killed tetracycline-resistant donors, N315 Δ cls2-tet^R, Nef Δ cls2-tet^R, or Nef pT181 was used. The former two carry a tetracycline resistance gene in the chromosome (*cls2* locus) while Nef pT181 carries a tetracycline resistance gene in plasmid pT181.

line 158: why now introducing sigH overexpression - may be remembered from the introduction but could aid the reader if shortly mentioned why and also how this is overexpressed.

R3-16 Transformation remained undetectable in Nef, and we just wanted to see whether nisin has effect when the transformation is detectable level (by SigH overexpression). But it is not essential for the main story. We deleted the description in main text, instead, briefly mentioning it in the figure caption.

Supplementary Figure 8 caption: Transformation remained undetectable, regardless of the Nisin treatment, and there were no significant effects in the SigH-overexpressing strain (Nefh).

line 155-158: This had to be explained much better - what is the assay exactly and what are you transforming with and what are you selecting for?

R3-17 Donor and recipient, together with transformation conditions, were indicated in each figure in the revised version. The Method section was also revised to separately explain the transformation assay using nisin

Line 523: To test the effect of nisin, purified plasmid was used as a donor (Supplementary Fig. 8). The 500 μ l of recipient cells from overnight cultures were washed and inoculated in 10 ml CS2 and grown for 2 h at 37°C with shaking. Then 8 μ g mL⁻¹ nisin was added to the medium and further grown for 8 h. Cells were then harvested and suspended in fresh 10 ml CS2 containing 10 μ g of purified pPHY-300PLK plasmid.

line 158: Compared to what? in supplementary figure 6 it appears that there is an effect of sigH but not of nisin.

R3-18 Nisin has no effect even in SigH-overexpressing cells. This was moved to a caption as mentioned above.

Figure 2c: Which type of DNA are you transforming with here, what are you selecting on and which strain is the recipient?

R3-19 We clarified this in the Figure.

line 166: This is very suspicious as your initial assumption was that *comG* is needed for competence and also the logic of the manuscript becomes obscure as all the optimizations of conditions (figure 1) were done with *comG* as reporter

R3-20 As explained above (R3-7), *PcomG* is the reasonable indicator of the SigH activity and expression of competence genes.

line 155-178: Where are the controls? There is a need for transformation with DNA lacking resistance genes and also genomic verification that the transformants have the recipient backbone but have acquired the transformed DNA.

R3-21 Control experiments were added as **Supplementary Fig. 7**.

To confirm that transformants were generated from recipient, Nef-pRIT5H (plasmid encoded *cm^R*) was used as the recipient. Heat-killed Nef-Delta-cls2 (chromosomal *tet^R*) was used as a donor. Selection is by tetracycline. All of the transformants carried both pRIT5H and *tet^R* gene, showing that the recipient (Nef-pRIT5H) acquired *tet^R* gene.

When the donor without tetracycline resistance (Nef) is used, no colony emerges under the selection by tetracycline (**Supplementary Fig. 7a bottom, 7b**).

line 184: What is the sigH expression level of the strain r59 compared to strains that are not transformable without artificial sigH overexpression?

R3-22 Thank you for this question. We tested the *PcomG* reporter expression in these strains. The r59 exhibited the reporter expression (**Supplementary Fig. 12**). Manuscript was revised as follow:

Line 222: *PcomG-gfp* reporter analysis indicated that r59 and s142 express GFP at frequencies of ~1% and ~0.1%, respectively, while s1587, r3, and r408 did not (**Supplementary Fig.12c**). Strains s142 and r3 became transformation competent after introducing a SigH-expressing plasmid (pRIT-sigH), suggesting that SigH expression is still a limiting step in the transformation of MSSA s142 and MRSA r3.

line 189: Also in these experiments are the controls missing.

R3-23 Negative control (Nef donor, without resistance gene) was added as **Supplementary Fig. 12b**.

line 204: You will need to show how the plates look with your transformants and also provide information on the number of transformants to demonstrate how well did cefmetazole work as selective agent.

R3-24 We have added the information of number of transformants (**Supplementary Table 5**) and

provided examples of plates (**Supplementary Fig. 13**). Following the first selection by cefmetazole, emergent colonies were replicated onto fresh plates containing the same concentration of cefmetazole and only those that could grow on replica were counted as transformants. This was added as **Supplementary Fig. 13**.

line 222-225: how do you control for the recipient being the correct strain? Transformants should be genome sequenced to demonstrate that they carry the backbone of the recipient in addition to the entire SCCMec cassette.

R3-25 Clonal complex typing (by Multiplex PCR: **Supplementary Fig. 17**) and genome sequencing (**Supplementary Fig. 16**) confirmed the genetic backbone of recipient and transformant.

line 228 and figure 3a: It is strange that you get weaker PCR bands for some of the transformants as you are not doing a quantitative PCR so the amount of template DNA should not be reflected in the PCR reaction.

R3-26 Although this is an end-point analysis of the PCR product, the template DNA was quantified and same quantities were used, the number of PCR cycles is moderate and the product quantity was not saturated (semi-quantitative PCR).

line 229: Very surprising you see equal transformability with coagulase negative staphylococci where there there should be restriction-modification barriers

R3-27 This would be the feature preferable for the interspecies transfer of SCC. Transforming DNA is taken up as the form of single strands and most restriction enzymes are not active against ssDNA. (In contrast, restriction system immediately targets incoming dsDNA in the case of phage transduction.) It would be only after conversion into dsDNA when the incorporated ssDNA becomes potentially sensitive to restriction. It is not known whether and how dsDNA converted after transformation is effectively protected in *Staphylococcus*. Interesting future study. We also previously reported that plasmids purified from *E. coli* HST04 *dam-/dcm-* can be used as donor for planktonic transformation (Nguyen *et al.*, 2018, ref#25).

line 236-241: Again you should provide genome sequence information rather than these fluffy PCR bands.

R3-28 Genome sequence confirmed the length of SCC and its chromosomal integration (**Supplementary Fig. 16**).

line 253: attB is not a commonly used term for SCCmec integration, is it the putative integration site in orfX? you need to reference where the attB sequence was described and provide the sequence.

R3-29 *attB* (attachment site) designates the specific site where SCC is integrated. The manuscript has been modified.

Line 55: ...integrated by Ccr (cassette chromosome recombinases) at a specific site (attachment site, *attB*) in *orfX* (a.k.a. *rmlH*, encoding rRNA methyltransferase¹¹) near the replication origin of the chromosome¹².

The *attB* locus contains 15 bp conserved core sequence (GAGGCGTATCATAAG). The sequence was indicated in **Supplementary Fig. 18**.

figure 4: Again, what was transformed with what?

R3-30 Information was added in the revised figure. We apologize for the inconvenience in the previous version.

Reviewers' Comments:

Reviewer #1:

Remarks to the Author:

Overall, I think the study was substantially improved. The authors performed many of the additional assays proposed that helped to better support their conclusions.

Still, in my opinion, the authors are in a point in which they did a series of interesting and scattered observations regarding the occurrence of transformation in *S. aureus*, SCCmec transfer, TCS and biofilms, many of them new, but which are not well integrated into a coherent hypothesis. The rationale of the different experiments are frequently not well explained and it is difficult to follow their logic and purpose. Also, it is challenging to understand what is the main conclusion that is supported by the different experiments performed. An overall revision of the main text having this in mind might help to focus and integrate the data. The effort might be worth trying given the amount of new data produced and the importance of the subject.

The major comments R1-1 to R1-6 were mainly addressed by the authors. Although the new experiments were performed to answer to the questions raised, the authors frequently failed to incorporate those experiments and their results in the main text of the manuscript. Most of these experiments and results were included as supplementary material, but the new results obtained were not integrated and concealed with previous data. This is the case for example of the experiments that studied the expression of comG in the presence of antibiotics (R1-26). The authors confirmed that the growth rate of strains was somehow affected in the presence of the antibiotics vancomycin and bacitracin, suggesting that the antibiotics have a general effect on strain growth and not an effect that was specific on comG expression. However, they did not change the main text to reflect that. Another case is the assays performed to answer question (R1-6) in which authors did co-localization studies for comG and comE. In this case the assays were neither integrated in the materials and methods nor in the main text.

The R1-7 was not totally addressed. Authors found that some of the donor strains actually carried prophages and plasmids by doing PCR assays. They end up stating that because they used heat-inactivated cells, that these cells cannot produce phage particles or conjugative machineries and for that reason SCCmec could have not been transduced or transferred by conjugation. Although I agree that conjugative machineries are probably not produced due to heat, phage particles could be present within cells and be preserved like the chromosomal DNA. There is thus the possibility that phages are involved in SCCmec transfer and this hypothesis cannot be totally discarded.

R1-11 – The limitations of the study are not thoroughly addressed. In particular, authors should address the following limitations: the fact that the condition in which the natural transformation of SCCmec was achieved are far from being similar to the real situation in staphylococcal ecological sites; the existence of conflicting results regarding comG function in transformation.

R1-24 – The use of comG as a reporter of transformation vs the finding that is not important for transformation is still not well-addressed. Authors say that comG is a reporter of the sigH activity, but this does not justify its use as a reporter of transformation when actually it seems not to be important for the function. This needs further clarification.

R1-30 - A sentence should be introduced in the text wherein the new results with the sigH null mutant in which no transformation was observed to occur in biofilms; should be concealed with the previous results in which the transformation frequency of Nefh (expressing sigH) and Nef were similar under biofilm-forming conditions.

R-39 – I continue to think that although authors showed that transformation does occur in biofilms, they cannot affirm this the most frequent mechanism of horizontal gene transfer of SCCmec. Plasmids could carry a large SCCmec like the type II within a conjugation event. In this case the size of the cassette is not a limitation for the occurrence of conjugation, like it is for transduction.

R1-42 – I continue to think that the frequencies of transformation in the planktonic and biofilm conditions cannot be comparable because were not done with same proportion of donor/recipient.

Authors included in the main text the proportions used in biofilm and planktonic assays. However, these values are not the correct ones. The ones that should be included are those corresponding to the CFU/mL of heat-inactivated cells over the CFU/mL of the recipient before transformation and not after transformation as indicated by the authors.

R1-43 – I continue to be confused regarding the mechanism associated to the transformation in biofilms. Is it similar or different from the one described for planktonic cells? Is it sigH dependent? comG-dependent, comE-dependent? TCS-dependent? Dependent on the recipient or donor or both? And how are these systems related? What is the mechanistic model that author envision given the results obtained so far?

Additional comment: The title should be changed to include the idea that MRSA can emerge by Natural genetic transformation, mainly in biofilms, not to be misleading. I think the words "can" and "biofilms" are important to be included because although authors showed that SCCmec can be transferred by natural transformation they were not able to prove in this study that this is the unique mechanism of transfer of this MGE. Also the frequency of transformation in biofilms appears to be much higher than that observed in planktonic cells and this is an important observation that is new and distinguishes this from previous studies.

Reviewer #2:

Remarks to the Author:

Review of the revised manuscript by Maree et al. The authors have addressed most of the points brought up by the reviewers in this revised version and thus I have no further queries. I think that the manuscript now fulfils all the requirements for a publication in Nature Communications.

Reviewer #3:

Remarks to the Author:

I acknowledge that the authors have done much to address the concerns that previously had been raised. However, I still find that the data are not convincing and does not allow the authors to state the title "MRSA emerges through natural genetic transformation". If the simple statement of the title was true all it would require would be an experiment where 10, 20 or 30 transformants are genome sequenced (entire genomes) to prove that they were all transformants and that the core genome of the recipient was still intact. That would avoid all the PCRs and make the story more convincing. Complicating the argument for the main claim of the manuscript is also the fact that the controls are scattered on a large number of figure, tables and supplementary figures of tables which actual makes its impossible to see if all has been controlled well for. I can here mention the response: "

The presence of donor DNA (tet, mecA ,or SCC) was confirmed by PCR (tet: Supplementary Fig. 7, mec: Fig. 3a, Supplementary Fig. 14, SCC: Fig. 3e,f,g). PCR (Supplementary Fig. 15) and genome sequencing (Supplementary Fig. 16) confirmed presence of SCCmec in the chromosome We also added a series of negative control experiments (listed below) confirming our experimental setting never detect spontaneous resistant mutant.

Fig. 2e: Dh, NefDcomE

Table 1, blue tone's cells: e.g. Nef (as donor that has no mecA), NefDcomE, 9sDcomE

Supplementary Fig. 7b: Nef (as donor with no tetracycline resistance gene)

Supplementary Fig. 11: NefDcomE

Supplementary Fig. 12: Nef (as donor with no tetracycline resistance gene)".

LAST but not least then the requested information concerning the actual number of transformants is now presented in supplementary table 5 and that reveals that in most cases the frequencies are based on less than 16 colonies on agar plates and n=1 or 2. The does not allow to state the title of manuscript and call for more simple and convinging data.

Major parts revised according to suggestions by editors and reviewers

We have revised the title to: “Natural transformation allows transfer of SCC_{mec}-mediated methicillin resistance in *Staphylococcus aureus* biofilms”. We have also toned-down descriptions throughout the manuscript.

We revised the manuscript and figures to increase the clarity (blue marker in revised manuscript). Major parts are as follows:

- Method sub-sections were restructured.
- The description of the screening for *mecA*-transformable strains was simplified in the main text, but an explanation was added to the supplementary material (Supplementary Table 2).
- The following text in the previous Discussion, which can be found in the referenced comprehensive review¹⁰, was omitted.

~~“In 1961, the first MRSA, which carried the Type I SCC_{mec}, was isolated in the United Kingdom while Types II and III, identified in the early 1980s in Japan and New Zealand, were clinically isolated and are reported as the largest types among SCCs^{30,31}. SCC_{mec} IV and V were described in United States and Australia but are relatively small and found primarily in community-acquired MRSAs^{32,33}. While Types I to V are dominant and widely distributed, diverse new variants have been reported (Types VI–XIII). The origins of SCC_{mec} are unclear but ancestral forms have been identified in coagulase-negative staphylococci such as *S. sciuri*, *S. fleuretti*, *S. xylosus*, *S. hominis*, and *M. caseolyticus*^{29,34,35}.”~~

- Many factors that were found to affect transformation in biofilm were summarized in the new Figure 5.
- The following description was simplified.

~~**Deleted:** “Moreover, it is also likely that the transformation process following competence gene expression is facilitated in biofilms because i) the P_{comG}-reporter expression did not increase in biofilm compared with planktonic condition (Fig. 2a: ~1% vs Fig. 1c: ~10%) but transformation was efficient in biofilms and ii) artificial SigH-overexpression did not increase transformation efficiency in biofilms (Fig. 2c). One possible explanation is better or more sustained cellular contact between donor and recipient cells in biofilm versus planktonic conditions even with a similar donor:recipient ratio.”~~

Instead, we simply mentioned in the legend of the new Figure 5: “Biofilms may also facilitate transformation due to better or more sustained cellular contact between donor and recipient cells,”

Point-to-point response to reviewers:

(specific parts changed are marked by yellow marker in revised manuscript)

Reviewer #1:

Overall, I think the study was substantially improved. The authors performed many of the additional assays proposed that helped to better support their conclusions.

Still, in my opinion, the authors are in a point in which they did a series of interesting and scattered observations regarding the occurrence of transformation in *S. aureus*, SCCmec transfer, TCS and biofilms, many of them new, but which are not well integrated into a coherent hypothesis. The rationale of the different experiments are frequently not well explained and it is difficult to follow their logic and purpose. Also, it is challenging to understand what is the main conclusion that is supported by the different experiments performed. An overall revision of the main text having this in mind might help to focus and integrate the data. The effort might be worth trying given the amount of new data produced and the importance of the subject.

We thank the reviewer for acknowledging the importance of our study. We have extensively revised the manuscript to address the reviewer's concerns as detailed below.

The major comments R1-1 to R1-6 were mainly addressed by the authors. Although the new experiments were performed to answer to the questions raised, the authors frequently failed to incorporate those experiments and their results in the main text of the manuscript. Most of these experiments and results were included as supplementary material, but the new results obtained were not integrated and concealed with previous data. This is the case for example of the experiments that studied the expression of *comG* in the presence of antibiotics (R1-26). The authors confirmed that the growth rate of strains was somehow affected in the presence of the antibiotics vancomycin and bacitracin, suggesting that the antibiotics have a general effect on strain growth and not an effect that was specific on *comG* expression. However, they did not change the main text to reflect that. Another case is the assays performed to answer question (R1-6) in which authors did co-localization studies for *comG* and *comE*. In this case the assays were neither integrated in the materials and methods nor in the main text.

We have now clarified and incorporated the following points into the revised manuscript:

General effect of antibiotics on P_{comG} expression

R1-26:

Line 173: "The antibiotics slightly affected cell growth (Supplementary Fig. 6a,b). Thus, the observed change in P_{comG-gfp} expression might be due to secondary effects of the antibiotics rather than their direct effects via TCSs. On the other hand, 8 or 16 µg mL⁻¹ nisin slightly, but reproducibly, increased reporter expression intensity in Nef-GFP (Supplementary Fig. 6c) though not in a statistically significant manner. No effect of nisin on cell growth was observed (Supplementary Fig. 6c)."

Co-localization of comG and comE expression

R1-6:

The explanation about the competence machinery genes was moved to the Introduction, and the co-localization experiments were added in the main body of the revised manuscript as follows:

Line 77: “namely the *comG* operon, encoding the pseudopilus that facilitates DNA access to the channel, and the *comE* operon, encoding an essential DNA internalization channel¹⁶.”

Line 109: “The *comG* operon promoter (P_{comG}) has been previously used to monitor SigH-dependent competence gene expression¹⁸. The promoters of the *comG* and *comE* operons (P_{comG} and P_{comE}) are both recognized by SigH¹⁸ and enhanced by ComK¹⁹, suggesting that the regulation of these two operons are under similar control by SigH and ComK. In order to verify whether the P_{comG} reporter is suitable to monitor the expression of competence machinery genes, we introduced the P_{comG} -*gfp* and P_{comE} -*dsRed* dual-fluorescence reporter plasmid into Nef. As expected, co-expression of these reporters was observed (Supplementary Fig. 1), suggesting that either of the promoters is suitable for monitoring competence gene expression.”

The R1-7 was not totally addressed. Authors found that some of the donor strains actually carried prophages and plasmids by doing PCR assays. They end up stating that because they used heat-inactivated cells, that these cells cannot produce phage particles or conjugative machineries and for that reason SCCmec could have not been transduced or transferred by conjugation. Although I agree that conjugative machineries are probably not produced due to heat, phage particles could be present within cells and be preserved like the chromosomal DNA. There is thus the possibility that phages are involved in SCCmec transfer and this hypothesis cannot be totally discarded.

Following the reviewer’s comment, we carefully revised the manuscript as follows:

Line 378: “In this study, we used heat-killed donors unable to synthesize conjugative machinery, eliminating the possibility of conjugation. Regarding transduction, heat may not completely destroy the infectious phage particles in donors and many of the tested donor strains had phages (Supplementary Fig. 19a). However, the finding that COLw/o ϕ (phageless) can serve as SCCmec donor (Fig. 3a) indicates independence from phage-dependent systems such as transduction. Importantly, the combination of COLw/o ϕ donor and Nef recipient resulted in no phages in the experimental system. We would like to also emphasize that, at least in the combination of heat-killed COLw/o ϕ (as SCCmec donor) with Nef or 9s recipients, the *comE* operon genes were essential for SCCmec transfer, clearly demonstrating that SCCmec was transferred by natural genetic transformation rather than phage-dependent systems or conjugation.”

R1-11 – The limitations of the study are not thoroughly addressed. In particular, authors should address the following limitations: the fact that the condition in which the natural transformation of SCCmec was achieved are far from being similar to the real situation in staphylococcal ecological sites; the existence of conflicting results regarding comG function in transformation.

We have addressed the above points in the text, which now reads:

Line 453: “One limitation of this study is that we did not examine natural conditions for MRSA emergence. It is unknown whether this occurs in some host organisms or certain abiotic environments. Further study is necessary but...”

We believe the non-essentiality of *comG* does not conflict with the results, and we explain this in the revised text:

Line 227: “Deletion of the *comG* operon (Nef Δ comG) severely impaired transformation in biofilms but did not abolish it completely. The non-essentiality of the *comG* operon encoding the pseudopilus is consistent with reports in model species¹⁶. In *Bacillus* spp., it is also known that the ComG pseudopilus is dispensable for DNA-binding by ComEA in the absence of a cell wall²⁷. Thus, under biofilm conditions where DNA availability and cell-to-cell contact are increased, it is conceivable that the requirement for the ComG-encoded pseudopilus is reduced.”

R1-24 – The use of *comG* as a reporter of transformation vs the finding that is not important for transformation is still not well-addressed. Authors say that *comG* is a reporter of the *sigH* activity, but this does not justify its use as a reporter of transformation when actually it seems not to be important for the function. This needs further clarification.

We have now addressed and clarified this point in the revised manuscript. 1) as stated above, the molecular functions of the *comG* and *comE* operon genes are now described in the Introduction. 2) The justification of the use of either the *comG* or *comE* reporter in this study is now added to the revised manuscript. 3) Misleading description that “*comG* is not necessary” is rephrased to properly state that “deletion of *comG* severely reduces the efficiency but does not abolish it”, and consistency with previous works was also added:

1)

Line 77: “namely the *comG* operon, encoding the pseudopilus that facilitates DNA access to the channel, and the *comE* operon, encoding an essential DNA internalization channel¹⁶.”

2)

Line 109: “The *comG* operon promoter (P_{comG}) has been previously used to monitor SigH-dependent competence gene expression¹⁸. The promoters of the *comG* and *comE* operons (P_{comG} and P_{comE}) are both recognized by SigH¹⁸ and enhanced by ComK¹⁹, suggesting that the regulation of these two operons are under similar control by SigH and ComK. In order to verify whether the P_{comG} reporter is suitable to monitor the expression of competence machinery genes, we introduced the P_{comG} -*gfp* and P_{comE} -*dsRed* dual-fluorescence reporter plasmid into Nef. As expected, co-expression of these reporters was observed (Supplementary Fig. 1), suggesting that either of the promoters is suitable for monitoring competence gene expression.”

3)

Line 224: “Transformation efficiencies of Nef increased up to day 3. No transformants could be detected in Δ H and Nef Δ comE, confirming that any transformation under biofilm conditions is dependent on SigH and the essential DNA channel encoded by the *comE* operon. Deletion of the *comG* operon (Nef Δ comG) severely impaired transformation in biofilms but did not abolish it

completely. The non-essentiality of the *comG* operon encoding the pseudopilus is consistent with reports in model species¹⁶. In *Bacillus* spp., it is also known that the ComG pseudopilus is dispensable for DNA-binding by ComEA in the absence of a cell wall²⁷. Thus, under biofilm conditions where DNA availability and cell-to-cell contact are increased, it is conceivable that the requirement for the ComG-encoded pseudopilus is reduced.”

R1-30 - A sentence should be introduced in the text wherein the new results with the sigH null mutant in which no transformation was observed to occur in biofilms; should be concealed with the previous results in which the transformation frequency of Nefh (expressing sigH) and Nef were similar under biofilm-forming conditions.

We modified the text as follows (please note that we changed the strain name from Nefh to Nef-H):

Line 207: “In Nef-H, the transformation frequency was similar to Nef under biofilm-forming conditions (Fig. 2c, left) while ΔH was non-transformable, indicating that SigH expression is essential but not the limiting factor for transformation in biofilm.”

R-39 – I continue to think that although authors showed that transformation does occur in biofilms, they cannot affirm this the most frequent mechanism of horizontal gene transfer of SCCmec. Plasmids could carry a large SCCmec like the type II within a conjugation event. In this case the size of the cassette is not a limitation for the occurrence of conjugation, like it is for transduction.

We agree with the reviewers and editors that we did not compare other transfer systems and toned-down the title according to recommendations. Some parts of the text were also revised accordingly. Also, we added references that imply possible roles of conjugation in SCC transmission as follows:

Title: **Natural transformation allows transfer of SCCmec-mediated methicillin resistance in *Staphylococcus aureus* biofilms**

Examples of text improvements (not all are included here)

Line 276: “In this study, we propose that natural transformation in biofilms ~~is the major mechanism for~~ mediates cell-to-cell transfer of SCCmecA based on the following experimental evidence.”

Line 376: “Nonetheless, the finding of *mecB*-carrying plasmid with conjugative elements in *S. aureus* suggests a possible role of conjugation in transferring SCC-carrying plasmid^{38,39}.”

Line 392: “we propose natural transformation as ~~the~~ a route for SCCmec transmission.”

R1-42 – I continue to think that the frequencies of transformation in the planktonic and biofilm conditions cannot be comparable because were not done with same proportion of donor/recipient. Authors included in the main text the proportions used in biofilm and planktonic assays. However, these values are not the correct ones. The ones that should be

included are those corresponding to the CFU/mL of heat-inactivated cells over the CFU/mL of the recipient before transformation and not after transformation as indicated by the authors.

We have corrected the information to indicate the donor to recipient ratio **before** the transformation assay, which is ~ 3 in both planktonic and biofilm conditions. The corrected text now reads:

Line 653: “The ratio of heat-killed donor (5×10^{10} CFU equivalent) to recipient (before transformation assay, $\sim 2 \times 10^{10}$ CFU) was approximately 2.5.”

Line 666: “The ratio of heat-killed donor ($\sim 2.5 \times 10^8$ CFU) to recipient (before transformation, $\sim 7.5 \times 10^7$ CFU) was approximately 3.”

Although the ratio of donor to recipient in both conditions is similar, we still avoided direct comparison between the two conditions as the contact efficiency between donor and recipient cells must be different and is higher in biofilms, as mentioned in the first revision. We believe there is no problem now.

R1-43 – I continue to be confused regarding the mechanism associated to the transformation in biofilms. Is it similar or different from the one described for planktonic cells? Is it *sigH* dependent? *comG*-dependent, *comE*-dependent? TCS-dependent? Dependent on the recipient or donor or both? And how are these systems related? What is the mechanistic model that author envision given the results obtained so far?

We have summarized our findings about the competence development in biofilms in the newly added Figure 5.

Figure 5 legend:

Figure 5. Factors affecting competence development in biofilms.

In CS2 medium, TCS13 and TCS17 are necessary for biofilm formation and transformation but can be compensated for by *sigH* expression. In MB medium biofilm, TCS13 and TCS17 are not mandatory for transformation. In CS2 medium, TCS7, TCS9, and TCS12 are not required for biofilm formation but do affect the expression of *com* genes. Biofilms may also facilitate transformation due to better or more sustained cellular contact between donor and recipient cells, reducing the necessity of the ComG pseudopilus, while ComE DNA internalization channels remain essential. Bacitracin and nisin affect natural transformation in biofilms, though detailed mechanisms remain elusive.

Line 414: “Based on our results, a regulatory framework for competence development is shown in Fig. 5”

Additional comment: The title should be changed to include the idea that MRSA can emerge by Natural genetic transformation, mainly in biofilms, not to be misleading. I think the words “can” and “biofilms” are important to be included because although authors showed that SCCmec can be transferred by natural transformation they were not able to prove in this

study that this is the unique mechanism of transfer of this MGE. Also the frequency of transformation in biofilms appears to be much higher than that observed in planktonic cells and this is an important observation that is new and distinguishes this from previous studies.

We thank the reviewers and the editor for this suggestion. We modified the title to the appropriate one.

Reviewer #2:

Review of the revised manuscript by Maree et al. The authors have addressed most of the points brought up by the reviewers in this revised version and thus I have no further queries. I think that the manuscript now fulfils all the requirements for a publication in Nature Communications.

We thank the reviewer again for the valuable suggestions about the roles of TCSs.

Reviewer #3:

I acknowledge that the authors have done much to address the concerns that previously had been raised. However, I still find that the data are not convincing and does not allow the authors to state the title "MRSA emerges through natural genetic transformation". If the simple statement of the title was true all it would require would be an experiment where 10, 20 or 30 transformants are genome sequenced (entire genomes) to prove that they were all transformants and that the core genome of the recipient was still intact. That would avoid all the PCRs and make the story more convincing. Complicating the argument for the main claim of the manuscript is also the fact that the controls are scattered on a large number of figures, tables and supplementary figures of tables which actually makes it impossible to see if all has been controlled well for. I can here mention the response: "

The presence of donor DNA (tet, mecA, or SCC) was confirmed by PCR (tet: Supplementary Fig. 7, mec: Fig. 3a, Supplementary Fig. 14, SCC: Fig. 3e,f,g). PCR (Supplementary Fig. 15) and genome sequencing (Supplementary Fig. 16) confirmed presence of SCCmec in the chromosome

We also added a series of negative control experiments (listed below) confirming our experimental setting never detects spontaneous resistant mutants.

Fig. 2e: Dh, NefDcomE

Table 1, blue tone's cells: e.g. Nef (as donor that has no mecA), NefDcomE, 9sDcomE

Supplementary Fig. 7b: Nef (as donor with no tetracycline resistance gene)

Supplementary Fig. 11: NefDcomE

Supplementary Fig. 12: Nef (as donor with no tetracycline resistance gene)".

LAST but not least then the requested information concerning the actual number of transformants is now presented in supplementary table 5 and that reveals that in most cases the frequencies are based on less than 16 colonies on agar plates and n=1 or 2. This does not allow to state the title of manuscript and call for more simple and convincing data.

We have toned-down the title and manuscript as recommended by reviewers and editors, and we avoided describing transformation as the sole major pathway.

We have also revised the manuscript with careful attention to mentioning and showing the controls in order to clarify the evidence for the natural transformation, and we hope it is now clear that the obtained transformants are not by spontaneous mutants or by other HGT mechanisms.

Reviewers' Comments:

Reviewer #1:

Remarks to the Author:

The authors have revised thoroughly the manuscript and tried to conceal all the results into a coherent hypothesis as previously suggested. And for that reason I have no further main remarks. I congratulate the authors for the huge amount of work performed and for the effort to respond to all Reviewers concerns. In the end I think the paper became more interesting and convincing.

Minor points:

In Introduction: *Staphylococcus sciuri* is nowadays considered to belong to a different genus (*Mammaliicoccus*). This needs to be added in lines 56-57.

In Discussion (lines 355): nowadays exist at least 13 different SCCmec types described.

Reviewer #3:

Remarks to the Author:

I appreciate that the authors have toned-down the claims of the manuscript. However, I still find that the actual number of colonies obtained by natural transformation (now presented in Supplementary table 3) is really very low. As this is the very basis of the story one must conclude that what is reported are rare events of which the biological importance may be limited.

Point-to-point response to reviewers' comments

Reviewer #1 (Remarks to the Author):

The authors have revised thoroughly the manuscript and tried to conceal all the results into a coherent hypothesis as previously suggested. And for that reason I have no further main remarks. I congratulate the authors for the huge amount of work performed and for the effort to respond to all Reviewers concerns. In the end I think the paper became more interesting and convincing.

We thank the reviewer for the valuable critique and feedback. We have further revised the manuscript to address the reviewer's minor points as detailed below.

Minor points:

In Introduction: *Staphylococcus sciuri* is nowadays considered to belong to a different genus (*Mammaliicoccus*). This needs to be added in lines 56-57.

We have added this information to the revised text which now reads:

Line 55: "homologues can be found in SCC, chromosomes, or plasmids in *Staphylococcus* (including *S. sciuri* that was recently reclassified to *Mammaliicoccus*¹⁰), or *Macrococcus* species¹¹."

In Discussion (lines 355): nowadays exist at least 13 different SCCmec types described.

We have added this information to the revised text which now reads:

Lines 350: "At least 13 different SCCmec types (I-XIII) have been described to date¹¹."

Reviewer #3 (Remarks to the Author):

I appreciate that the authors have toned-down the claims of the manuscript. However, I still find that the actual number of colonies obtained by natural transformation (now presented in Supplementary table 3) is really very low. As this is the very basis of the story one must conclude that what is reported are rare events of which the biological importance may be limited.

To address the reviewer's concern, we have revised the manuscript to avoid the description of "efficient natural transformation" in the main text:

Lines 95: "Furthermore, we present experimental evidence of inter- and intraspecies transfer of *SCCmec* between staphylococcal cells via natural transformation."

Line 193: "Biofilm growth conditions enhance natural transformation efficiency"